METHODS

# Mathematical basis and toolchain for hierarchical optimization of biochemical networks

**Nisha Ann Viswan[1,2], Alexandre Tribut[3,4], Manvel Gasparyan[3], Ovidiu Radulescu[3]\*, Upinder S. Bhalla**[1]\*

**1** National Centre for Biological Sciences, Tata Institute of Fundamental Research, Bangalore, India, **2** The University of Trans-Disciplinary Health Sciences and Technology, Bangalore, India, **3** Laboratory of Pathogens and Host Immunity, University of Montpellier, CNRS and INSERM, Montpellier, France, **4** Ecole Centrale de Nantes, Nantes, France

\* ovidiu.radulescu@umontpellier.fr (OR); bhalla@ncbs.res.in (USB)

**Data Availability Statement:** HOSS code: https://github.com/BhallaLab/HOSS, FindSim code:

## Abstract

Biological signalling systems are complex, and efforts to build mechanistic models must confront a huge parameter space, indirect and sparse data, and frequently encounter multiscale and multiphysics phenomena. We present HOSS, a framework for Hierarchical Optimization of Systems Simulations, to address such problems. HOSS operates by breaking down extensive systems models into individual pathway blocks organized in a nested hierarchy. At the first level, dependencies are solely on signalling inputs, and subsequent levels rely only on the preceding ones. We demonstrate that each independent pathway in every level can be efficiently optimized. Once optimized, its parameters are held constant while the pathway serves as input for succeeding levels. We develop an algorithmic approach to identify the necessary nested hierarchies for the application of HOSS in any given biochemical network. Furthermore, we devise two parallelizable variants that generate numerous model instances using stochastic scrambling of parameters during initial and intermediate stages of optimization. Our results indicate that these variants produce superior models and offer an estimate of solution degeneracy. Additionally, we showcase the effectiveness of the optimization methods for both abstracted, event-based simulations and ODE-based models.

## Author summary

Biochemical pathway models integrate quantitative and qualitative data to understand cell functioning, disease effects, and to test treatments in silico. Constructing and optimizing these models is challenging due to the complexity and multitude of variables and parameters involved. Although hundreds of biochemical models have been developed and are available in repositories, they are rarely reused. To enhance the utilization of these models in biomedicine, we propose HOSS, an innovative hierarchical model optimization method. HOSS takes advantage of the modular structure of pathway models by breaking

https://github.com/BhallaLab/FindSim, HillTau
simulator: https://github.com/BhallaLab/HillTau,
MOOSE simulator: https://github.com/BhallaLab/
moose-core, Figure generation scripts: https://
github.com/BhallaLab/hossFigs HiNetDecom code:
https://github.com/Computational-Systems-
Biology-LPHI/HiNetDecom.

**Funding:** • This study was supported by the Centre
Franco-Indien pour la Promotion de la Recherche
Avancée (CEFIPRA) grant 68T08-3 to OR and USB,
by a fellowship award, number DBT/2018/NCBS/
998 to NAV from the Department of Biotechnology,
Government of India, and by institutional funding
to NCBS-TIFR under DAE Project Identification No.
RTI 4006 to USB to USB from the Department of
Atomic Energy, Government of India. The funders
did not play any role in the study design, data
collection and analysis, decision to publish, or
preparation of the manuscript.

**Competing interests:** The authors have declared
that no competing interests exist.

down large mechanistic computational models into smaller modules. These modules are then optimized progressively, starting with input modules and following causality paths. This method significantly reduces the computational burden as each step involves solving a simpler problem. By making the optimization process more manageable, HOSS accelerates the lifecycle of biochemical models and promotes their broader use in biomedical research and applications.

## Introduction

Many large biochemical pathway models have been developed since the early days of systems biology. These models take many different formalisms, including visual representations of data, such as protein interaction networks [1], and executable models like chemical reaction networks which can be solved with ordinary differential equations [2], or using stochastic calculations [3], boolean models [4], and more recently, HillTau abstractions [5]. Among executable models, ODEs provide accurate representation of pathway dynamics, but incorporate many unknown parameters.

Two key advances have opened up the possibility of scaling up systems models substantially, in terms of complexity and reproducibility. First, there is now a rich ecosystem of data resources and data mining resources, both from structured databases and from the much broader but unstructured scientific literature. These approaches have already been used to scale up pathway diagrams and interaction networks [6]. Second, the advent of numerous high-throughput methods such as phosphoproteomics, imaging, and mass spectrometry promise far larger and internally consistent datasets (see [7]) than the extant patchwork of precise but once-off biochemical experiments performed by individual laboratories.

However, in spite of a few attempts [8, 9], the model development process is far from being automatic and standardized. Parameter optimization frameworks have been implemented in a diverse manner, with different specification formats for the parameters, the experimental datasets, the parameter bounds, the objective functions and the choice of optimization methods [10–23]. This is in part due to the very wide diversity of experimental inputs used to constrain such models, but also due to the inherent contradictions and sparseness of the data. For example, to compensate for the sparseness of specific datasets and further constrain the model, it is not uncommon to amalgamate the findings from various publications. These data sources may utilize experimental preparations that differ significantly or even involve different classes of organisms [2]. This practice introduces inconsistent experimental inputs into the model. Consequently, model development is highly idiosyncratic, and different modelers may arrive at quite distinct models or parameter sets despite drawing on similar data sources. Our framework systematizes how experiments are used for optimization, and provides a way to assign a numerical confidence in each, thus ensuring that optimization runs are reproducible and can evolve as better experiments become available.

There have been previous ambitious efforts to systematically funnel many experimental inputs into detailed and biologically driven models [24]. Such efforts require the integration of large-scale systematic data gathering with data management and modelling (e.g, SPEDRE [25]). The current paper focuses on standardizing the calibration and optimization stages of model development, given a large but sparse set of experimental data. We build on our recently developed framework (FindSim [26]) for curating a very wide range of biochemical and physiological experiments, representing it in a consistent format, and using such curated experiment definitions to drive multiscale models. In principle, each new experiment should

improve our understanding of biological systems, and thus help us to refine models of these systems. This amounts to a multi-parameter optimization problem. Its result should be a model that fits experiments as well as possible within the limitations of the model, while incorporating expert evaluation of the relative reliability of different experiments.

We formalize and implement a general methodology for solving multi-parameter optimization problems by leveraging the modularity property of biochemical networks. These networks consist of groups of species and biochemical reactions that function autonomously. In this work, we present innovative algorithmic approaches for systematically performing modular decomposition of biochemical networks, described by various methods including ordinary differential equations and event-based modelling. Additionally, we connect the modular approach to hierarchical optimization, offering fully automated methods to handle data and models in a hierarchical manner.

Inspired by game theory and now with multiple applications in science and engineering, hierarchical optimization decomposes a complex optimization problem into several coupled simpler problems [27, 28]. Although NP-hard in general, hierarchical optimization becomes easier for nested hierarchies, where lower levels depend on fewer parameters than the upper levels. We refer to such a method as nested hierarchical optimization and provide algorithmic solutions for implementing it in pathways.

We report the development of an optimization pipeline, HOSS, implementing nested hierarchical optimization. We illustrate its use on an extant database of over 100 experiment definitions in the domain of synaptic signalling' encoded into the FindSim format, to improve the parameterization of a set of models of major signalling pathways involved in synaptic signalling and cell proliferation. HOSS utilizes FindSim [26] in order to consistently evaluate models based on a specified set of experiments.

We show how our hierarchical approach addresses many of the challenges of parameter optimization problems, and outperforms a *flat* (i.e., non hierarchical single stage) optimization approach in efficiency, structure, and accuracy.

Our pipeline implements tools and standard formats for automatically handling models, data and machine learning (ML) scenarios. All our models are encoded using the Systems Biology Markup Language (SBML), a well-established format in Systems Biology. Both data and optimization choices, including flat and hierarchical optimization, with the definition of sub-models in the hierarchical case, are encoded using JavaScript Object Notation (JSON) files. The goal is to make ML more reproducible and accessible to non-experts, while increasing productivity for experts. This situates our effort within the field of Automated Machine Learning (AutoML), a relatively new area that makes advanced ML techniques more accessible and accelerates research processes in computational biology [29].

Decomposition of optimization problems has been previously explored for hybrid Petri Net models of signalling [30], using autonomous modules, though without extending the approach to networks with feedback. In cases involving sparse data, under more restrictive conditions, the decomposition of objective functions into a sum of independent terms has been proposed as a way to split optimization into simpler independent problems [31]. Decomposition in the presence of feedback has also been addressed through the method of dependent inputs [32]. Hierarchical optimization has been applied to the optimization of systems biology models, particularly in its simplest form, known as bilevel optimization [33, 34]. In this paper, we present a comprehensive mathematical and algorithmic solution for the hierarchical decomposition and multilevel hierarchical optimization of general networks, including feedback, and implement it in software. Our toolset employs JSON format specifications for both optimization configuration and experimental datasets, ensuring easy interoperability and reusability, even across toolsets with different syntaxes.

## Methods

### Mathematical formalism

**Objective (cost) function.** A popular choice of objective function is the log likelihood. For data with normally distributed deviations, it reads:

$$L(\theta, s, \sigma) = \frac{1}{2} \sum_{i=1}^{N} \left[ \log(2\pi\sigma_i^2) + \left( \frac{y_i - s_i x_i(\theta)}{\sigma_i} \right)^2 \right], \qquad (1)$$

where $y_i$ and $x_i(\theta)$ are observed and predicted concentrations of the $i^{\text{th}}$ observed species, respectively, and $\theta$ are kinetic parameters. Parameters $s_i$ are scaling parameters, accounting for the fact that the measurements are not absolute and $\sigma_i$ are standard deviation parameters (see [33]).

In the HOSS calculations we perform two levels of scoring. First, for each experiment for which a model is tested, we obtain a normalized root-mean-square cost similar to the above calculation, except it is normalized to the maximum of the experiment readout for molecule $y_i$ among all the observations of the same variable at different times or in different conditions:

$$\text{NRMS}(\theta) = \sqrt{\frac{1}{N_d} \sum_{i,k} \left( \frac{y_{ik} - x_i(t_k, \theta)}{m_i} \right)^2}, \qquad (2)$$

where $m_i$ is the maximum value of the observed variable $y_i$ and $x_i(t_k, \theta)$ is its predicted value at the time $t_k$, and $N_d$ is the number of data points (terms in the sum).

To handle multiple data sets and multi-objective optimization we adopt a weighted sum approach. We define the weighted normalized cost, that combines values of (2) obtained in multiple datasets:

$$\text{WNRMS}(\theta) = \sqrt{\frac{\sum_j w_j (\text{NRMS}_j(\theta))^2}{\sum_j w_j}}, \qquad (3)$$

where $w_j$, and $\text{NRMS}_j$ are positive weights, and normalized root mean costs of individual datasets, respectively.

We used this simple way to rescale the terms in the least-squares objective function, rather than the log likelihood (1), because the highly heterogeneous nature of the experimental datasets does not lend itself to mathematically expressed statistical quantification of uncertainty. Specifically, different experiments disagree and are performed under different biological contexts. While some experiments do provide error bars, there is no rigorous way to combine such estimates when the main source of difference between readings is not measurement noise, but details of the experimental system being employed. We instead chose to incorporate a semi-quantitative expert assessment of data reliability by assigning weights to each experiment used for estimating the cost function value. Our experimental data format (FindSim format, [26]) includes error estimates for future use of more elaborate objective functions in cases where datasets are more homogeneous.

**Flat and hierarchical optimization.** Parameter optimization involves minimization of an objective function $f : C \subset \mathbb{R}^n \to \mathbb{R}$, where $C$ is a space of constraints, $\boldsymbol{p} \in C$ a vector of parameters. The flat method consists of solving the problem:

$$\min_{\boldsymbol{p} \in C} f(\boldsymbol{p}). \qquad (4)$$

There are many methods to solve (4). In our framework we use multistart optimization, by launching local search procedures from randomly chosen starting points generated uniformly in logarithmic scale:

$$\boldsymbol{p} = \tilde{\boldsymbol{p}} \exp(\log(\boldsymbol{a}) + \log(\boldsymbol{b}/\boldsymbol{a})\boldsymbol{U}), \tag{5}$$

where $\tilde{\boldsymbol{p}}$ is a nominal guess, $\boldsymbol{U} = (U_1, U_2, \ldots, U_n)$ a vector of random, independent variables whose distribution is uniform over [0, 1] or standard normal, $\boldsymbol{a} = (a_1, a_2, \ldots, a_n)$ and $\boldsymbol{b} = (b_1, b_2, \ldots, b_n)$ are vectors of positive scales, such that $0 < a_i < 1 < b_i$ for $1 \le i \le n$. All the vector multiplications in (5) are elementwise. By using this procedure, the range of start parameters is from $\bar{p}_i a_i$ to $\bar{p}_i b_i$.

We refer to this procedure as *parameter scrambling*. Despite its simplicity, multistart optimization with logarithmic sampling has proven to be effective in benchmarks of biochemical pathways [35]. Similar to [35], which uses a logarithmic scale for both initial and evolving parameter values, we allow an option between logarithmic and linear scales for the optimizer, with logarithmic as the default.

In hierarchical optimization [27], $K$ sub-problems, each one defined by an objective function $f_i : C \to \mathbb{R}$, $i = 0, \ldots, K - 1$, are solved iteratively. Parameters of the problem are grouped in $K$ groups $\boldsymbol{p} = (p_0, \ldots, p_{K-1})$, where $p_i \in \mathbb{R}^{n_i}$ for $0 \le i \le K - 1$ and $n = n_0 + \ldots + n_{K-1}$ is the total number of parameters. We look for $\boldsymbol{p}^* = (p_0^*, \ldots, p_{K-1}^*) \in C \subset \mathbb{R}^n$, solution of:

$$
\begin{aligned}
&\min_{p_{K-1}} f_{K-1}(p_0^*, \ldots, p_{K-2}^*, p_{K-1}) && \text{where } p_{K-2}^* \text{ solves} \\
&\min_{p_{K-2}} f_{K-2}(p_0^*, \ldots, p_{K-3}^*, p_{K-2}, p_{K-1}) && \text{where } p_{K-3}^* \text{ solves} \\
&\quad\quad\vdots && \quad\quad\vdots \\
&\min_{p_1} f_1(p_0^*, p_1, \ldots, p_{K-1}) && \text{where } p_0^* \text{ solves} \\
&\min_{p_0} f_0(p_0, p_1, \ldots, p_{K-1}) && p \in C.
\end{aligned}
\tag{6}
$$

The case $K = 2$ is known as bilevel optimization [28]. In this case the optimization of $f_0$ is called lower-level problem, whereas the optimization of $f_1$ is the upper-level problem.

The problem (6) is difficult, because each individual problem has to be solved for multiple values of the remaining variables, all subjected to the constraints $C$. Indeed, it has been proved that even apparently simple bilevel optimization problems are NP-hard [28]. However, the solution of bilevel optimization is straightforward if the lower-level problem has unique analytic solution. In this case, the naive algorithm, utilizes the solution of the lower-level problem to eliminate $p_0$ and reduce the upper-level optimization to minimizing a composed function that depends on $p_1$ only, is effective. Bilevel optimization with analytic solution for the lower-level problem has already been used for systems biology models. In this case the lower-level parameters are the scaling and standard deviation parameters $s_i, \sigma_i$ introduced in (1), that can be optimized by analytic formulas, see [33].

Another simple hierarchical optimization case is when the functions $f_i, 0 \le i \le K - 1$ depend on nested sets of parameters and the set of constraints factorizes $C = C_0 \times C_1 \times \ldots \times$

$C_{K-1}$. Then, (6) reads:

$$
\begin{aligned}
&\min_{p_{K-1} \in C_{K-1}} f_{K-1}(p_0^*, \ldots, p_{K-2}^*, p_{K-1}) && \text{where } p_{K-2}^* \text{ solves} \\[2mm]
&\min_{p_{K-2} \in C_{K-2}} f_{K-2}(p_0^*, \ldots, p_{K-3}^*, p_{K-2}) && \text{where } p_{K-3}^* \text{ solves} \\[2mm]
&\qquad\qquad \vdots && \qquad \vdots \\[2mm]
&\min_{p_1 \in C_2} f_1(p_0^*, p_1) && \text{where } p_0^* \text{ solves} \\[2mm]
&\min_{p_0 \in C_0} f_0(p_0)
\end{aligned}
\tag{7}
$$

We call the problem (7), **nested hierarchical optimization**. Nested hierarchical optimization can be solved iteratively, starting with the last problem in (7).

**Nested hierarchical decompositions.** A biochemical model is defined by a set of reactions $\mathcal{R}$ and a set of species $\mathcal{S}$. We also define the stoichiometric matrix $\boldsymbol{S}$, whose elements $S_{ij}$ represent the number of molecules of the species $i$ produced (if $S_{ij} > 0$) or consumed (if $S_{ij} < 0$) by the reaction $j$. Furthermore, the reaction rate vector $\boldsymbol{R}(\boldsymbol{x}, \boldsymbol{p}) = (R_1(\boldsymbol{x}, \boldsymbol{p}), \ldots, R_r(\boldsymbol{x}, \boldsymbol{p}))$ is a function of species concentrations $\boldsymbol{x} = (x_1, \ldots, x_N)$ and kinetic parameters $\boldsymbol{p}$. Each reaction $j$ is characterized by a parameter vector $\boldsymbol{p}_j$, therefore we have $\boldsymbol{p} = (\boldsymbol{p}_1, \ldots, \boldsymbol{p}_r)$.

Species concentrations evolve in time as a result of chemical reactions. These define a semiflow (time dependent mapping of the species concentrations, enabling the computation of future concentrations based on the present ones) $\phi(t, \boldsymbol{x}; \boldsymbol{p}), t \geq 0$ such that $\boldsymbol{x}(t) = \phi(t, \boldsymbol{x}_0; \boldsymbol{p})$ represents the species concentration vector starting from initial values $\boldsymbol{x}(0) = \boldsymbol{x}_0$. The semiflow results from the integration of ODEs in chemical kinetics models or from the simulation of event driven dynamics in HillTau abstractions [5].

Some species forming a subset $BS \subset \mathcal{S}$ are buffered, and their concentrations are kept constant.

Our construction relies on the following concept. We call **autonomous pair**, a pair of reaction and species subsets $(I, J), I \subset \mathcal{S}, J \subset \mathcal{R}$ that satisfy:

1. if a species is in the subset $I$, then all the reactions consuming or producing this species are in the corresponding reaction subset $J$, namely if $i \in I$ then $j \in J$ whenever $S_{ij} \neq 0$.

2. if a reaction is in the subset $J$, then all the species on which the reaction rate depends are in the corresponding species subset $I$, unless these species are buffered, i.e. if $j \in J$ then $i \in I$ whenever $\frac{\partial R_j}{\partial x_i} \neq 0$ and $i \notin BS$.

Let $\boldsymbol{x}_I$ be the concentration vector of the species in $I$ and $\boldsymbol{p}_J$ the kinetic constants of the reactions in $J$. From the above definition it follows that $\boldsymbol{x}_I$ can be computed at any positive time $t$ by a semiflow depending only on the parameters $\boldsymbol{p}_J$, namely $\boldsymbol{x}_I(t) = \phi_I(t, \boldsymbol{x}_I(0); \boldsymbol{p}_J)$. Consider the data subset $D_I$, consisting of observations $\boldsymbol{y}_I$ of the species $\boldsymbol{x}_I$ only. Then, the objective function measuring the difference between observed and predicted values of $\boldsymbol{x}_I$ depends only on $\boldsymbol{p}_J$, namely

$$
f_J(\boldsymbol{p}_J) = \sum_{i \in I} \sum_k (y_{ik} - \phi_i(t_k, \boldsymbol{x}_I(0); \boldsymbol{p}_J))^2.
\tag{8}
$$

Suppose now that we find species and reaction subsets,

$$I_0 \subset I_1 \subset \ldots \subset I_{K-1} \quad = \mathcal{S},$$

$$J_0 \subset J_1 \subset \ldots \subset J_{K-1} \quad = \mathcal{R},$$

(9)

such that $(I_k, J_k)$ are autonomous pairs for all $0 \leq k \leq K-1$.

We call (9) a **nested hierarchical decomposition**. Optimization of the objective functions (8) can then be done hierarchically, as in (7).

**Constructing nested hierarchical decompositions using the interaction graph.** Let us define the interaction digraph as $\mathcal{I} = (V, E)$, where $V$ is the set of vertices (all species) and $E$ is the set of edges. A pair of species $(j, i) \in E$ defines an edge from $j$ to $i$ if and only if there is a reaction that consumes or produces the species $i$, and its rate depends on the concentration of the species $j$. This graph is used to define causality relations between species, namely we say that $j$ is causal to $i$, $j \rightsquigarrow i$, if $j$ is connected to $i$ by a path in $\mathcal{I}$. All the species $j$ causal to $i$ are needed for computing the time evolution of $x_i$.

Strongly connected components (SCC) of $\mathcal{I}$ are subsets $K \subset V$ such that $j \rightsquigarrow i$ and $i \rightsquigarrow j$ for all $i, j \in K$, maximal with respect to this property. In this paper, we refer to SCCs as **blocks**. Blocks form a partition of the species set $I$. This partition can be used to define a SCC quotient graph as follows: blocks are vertices of the SCC quotient graph, and two blocks are connected if there is one species in one block connected in the interaction graph to a species in the other block. The SCC quotient graph is always acyclic (see Fig 1 and [36]).

The following property is important for building nested hierarchical decompositions.

**Property:** For any block $K$ and any subset $I$ of an autonomous pair $(I, J)$, one has either $K \subset I$ or $K \cap I = \emptyset$.

Thus, we can build a nested hierarchical decomposition by using the blocks and the quotient graph. The lowest level subset $I_0$ is the union of blocks that are roots of the quotient graph, i.e. blocks having no incoming connections. The corresponding reaction subset $J_0$ is made of all reactions producing or consuming species from $I_0$. The next level $I_1$ is obtained by adding to the roots all the blocks receiving direct connections only from the roots, and so on and so forth. Algorithmically, one must associate a hierarchical level $l$ to each block, defined as the length of the longest path from the roots to the block (see Fig 1). Then, the set $I_l$ is the union of all blocks with a hierarchical level smaller than $l$ (see Fig 1).

Although the nested hierarchical decomposition (9) is not unique, the decomposition obtained by this procedure is unique and has the advantage of minimality. More precisely, $I_0$ is the minimal subset containing the root blocks, such that $(I_0, J_0)$ is autonomous. $I_1$ is the minimal subset containing the species $I_0$ and all the species receiving direct interactions only from $I_0$.

The quotient graph also provides a useful data structure for parallel optimization of the parameters. Thus, each tree originating from a root corresponds to terms in the objective function that can be optimized independently of the others.

**Hierarchical decompositions with feedback.** In some signalling pathway models, downstream molecules regulate upstream ones through feedback [37]. This can result in all species influencing each other, forming a single block where hierarchical and flat optimizations are equivalent. However, even in these cases, we can identify smaller blocks and decompose the network hierarchically (Figs 2 and 3).

Some blocks are no longer autonomous in the presence of feedback, because they receive input from higher level blocks. In order to compute the ODE solutions and objective functions, we use a standard approach known as *dependent input* in systems biology [32]: the time dependent signal coming from higher level blocks is replaced by experimental data.

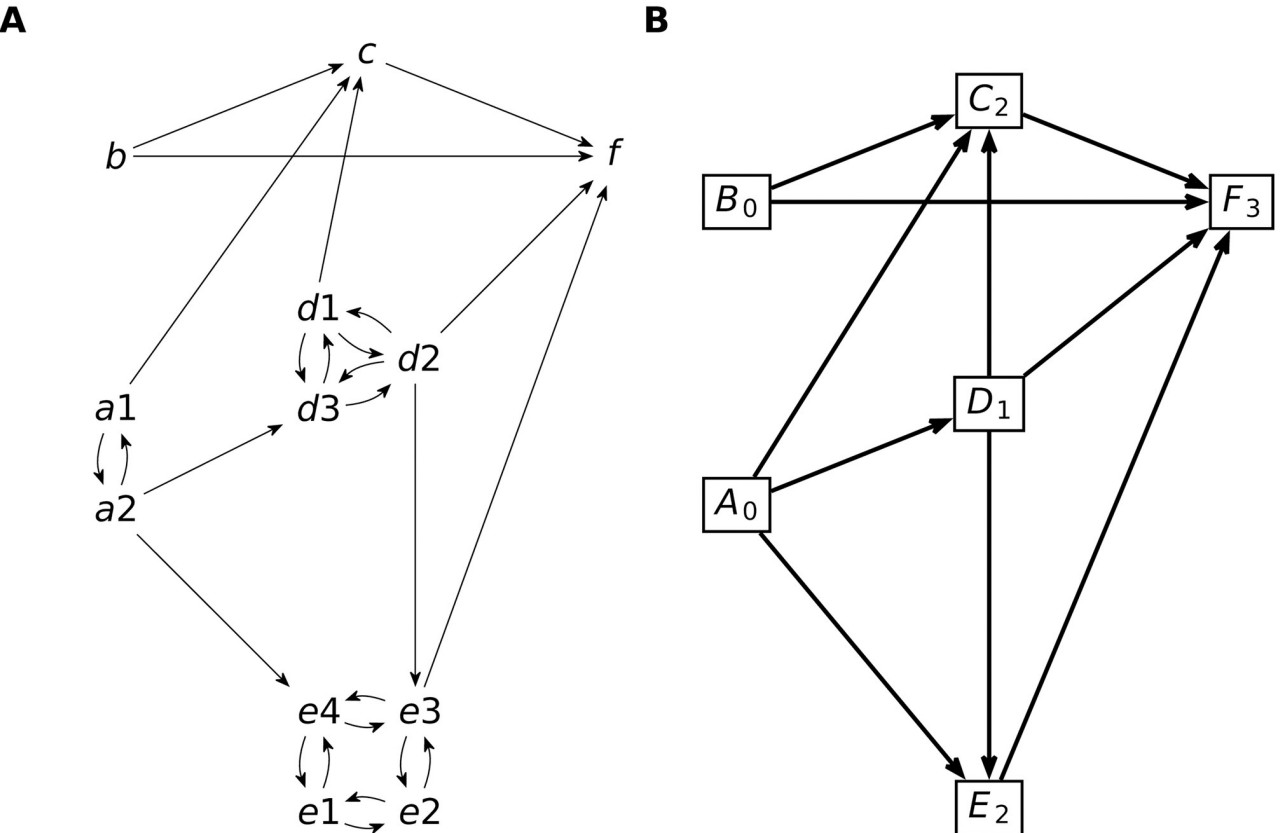

**Fig 1. Interaction and quotient graphs used for the hierarchical decomposition.** A) The interaction graph is a directed graph whose nodes are biochemical species. One source species acts on a target species if there is a reaction consuming or producing the target, whose rate depends on the concentration of the source. In this example, all species are considered self-causal; however, for simplicity, self-interactions are not shown. The strongly connected components (SCC) are maximal sets of nodes such that there are paths connecting each node to any other node. This graph has six SCCs: $A_0$ = {$a_1, a_2$}, $B_0$ = {$b$}, $D_1$ = {$d_1, d_2, d_3$}, $C_2$ = {$c$}, $E_2$ = {$e_1, e_2, e_3, e_4$}, $F_3$ = {$f$}. B) The quotient graph is an acyclic directed graph, whose vertices are the SCC of the interaction graph. Two SCC are connected in the quotient graph if there is a species in one connected to a species in the other. The hierarchy level of a block (SCC) is the length of the longest path in the quotient graph, connecting a root to the block. In this example, blocks $A$, $B$ are roots and have level 0, block $D$ has level 1, blocks $C$, $E$ have level 2 and $F$ have level 3. The corresponding autonomous subsets are defined as $I_0 = A_0 \cup B_0$, $I_1 = A_0 \cup B_0 \cup D_1$, $I_2 = A_0 \cup B_0 \cup D_1 \cup C_2 \cup E_2$, $I_3 = A_0 \cup B_0 \cup D_1 \cup C_2 \cup E_2 \cup F_3$.

We summarize the hierarchical decomposition procedure in the case with feedback, leaving the details to a separate paper.

In the presence of feedback, two concepts are key for the hierarchical decomposition. The first concept is *r-causality*. A species $j$ is r-causal to a species $i$, $j \xrightarrow{r} i$, if $j$ is connected to $i$ by a path in the interaction graph $\mathcal{I}$, of length smaller than or equal to $r$. The introduction of r-causality imposes an upper limit on the length of paths connecting species in the interaction graph. By taking the value of $r$ sufficiently large one can thus break feedback loops.

The other concept is *agony*, a measure used to quantify the hierarchical organization of directed networks [38, 39]. It helps in identifying and evaluating the hierarchical structure within a network by penalizing the inconsistencies present in the hierarchy. More precisely, integer variables representing the level in the hierarchy are associated to each species in the network. Then agony is a function of all these levels and of the interaction graph, that penalizes the edges for a node with high level to a node with lower level. The levels that minimize agony are then used to define the hierarchy.

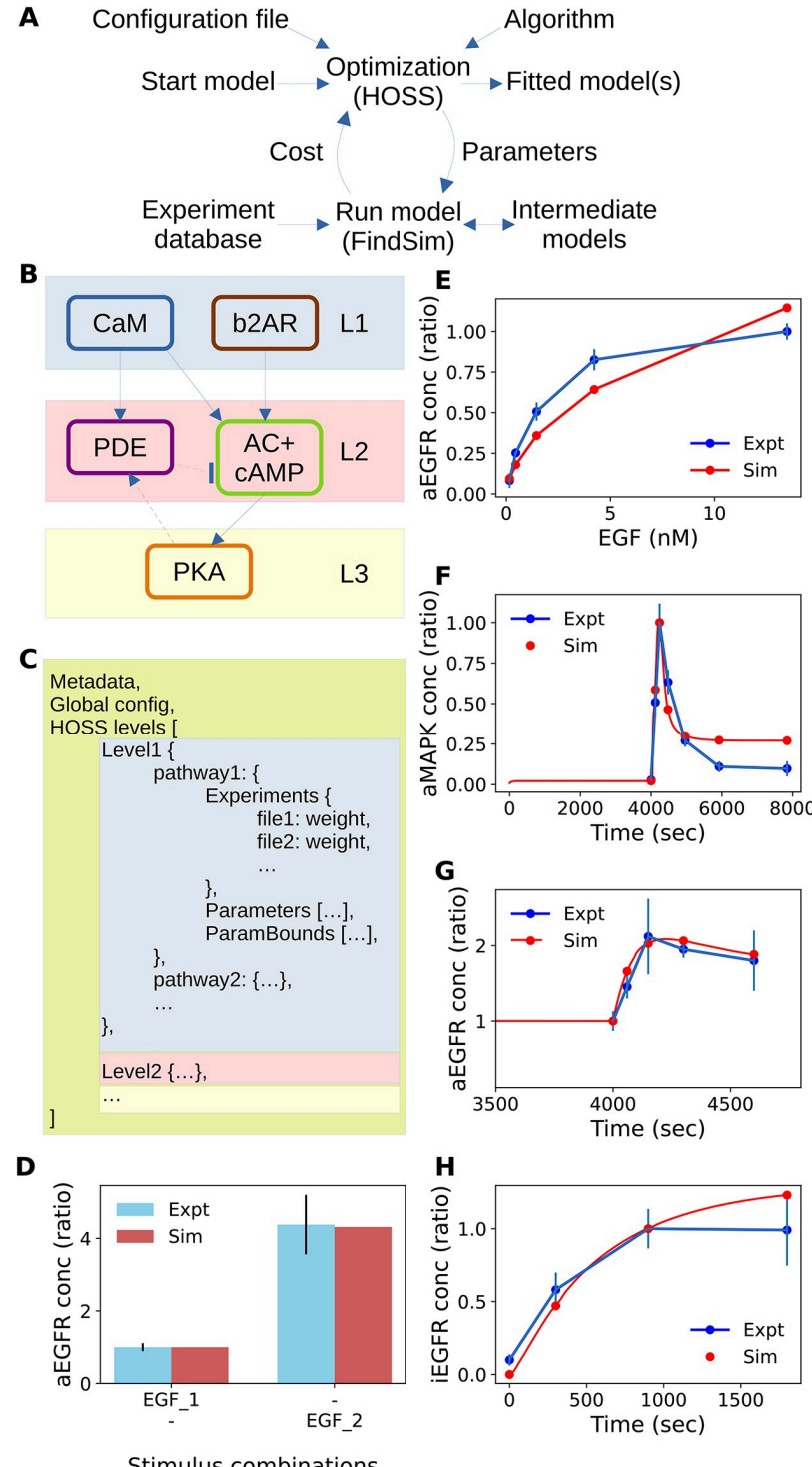

**Fig 2. Optimization framework.** A: Schematic of optimization pipeline. HOSS follows a pipeline defined in a JSON configuration file. HOSS orchestrates the operations of FindSim which takes a model, modifies its parameters, runs the model against specified experiments, and returns a cost representing the distance between data and model predictions. This cost is used by the algorithm. B: Typical model decomposition into levels. L1 depends only on inputs, L2 depends only on L1 and inputs, and L3 depends on all upstream pathways. Within a level we can have multiple signalling blocks provided they do not depend on each other. However, we may have cross-interactions or feedback (arrows with dashed lines), which may require the pipeline to repeat one or more levels. C: pseudocode for definition of the HOSS pipeline. Within each level we can have multiple pathways, each of which needs a list of experiments, parameters and

optionally parameter bounds. Colors map to corresponding levels of the model from panel B. D—H: Examples of experiments from database run using FindSim to obtain cost function values. D: Bar chart. EGF is provided at baseline level (0.1 nM, named EGF_1) and at stimulus level (1.5625 nM, named EGF_2), and the resultant level of activated EGF receptor (aEGFR) is found. E: Dose-response. EGF is provided at a series of fixed input levels, and the steady-state levels of aEGFR are measured F: Time-series. A 7.8125 nM step stimulus of EGF is applied at t = 4000s, and the level of activated MAPK is read out. G: Time-series. At 4000 seconds, after settling, EGF is raised from 0.5 nM to 1.5625 nM, and level of aEGFR is read out. H: Time-series. EGF is set to 0.15625 nM from t = 0, and level of internalized EGFR is read out.

Our procedure to compute the hierarchical decomposition of a network with feedback is as follows:

- First define r-blocks, such as maximal subsets such that any species is r-causal to any other.

- Because r-causality is not an equivalence relation, r-blocks can overlap. Generate a consolidated r-block partition (also named r-SCC partition) by agglutinating r-blocks that overlap.

- Use the r-SCC partition to define a r-quotient graph in the same way as the quotient graph was defined from the SCC partition. The nodes of the r-quotient graph are the consolidated r-blocks.

- Use agony to define hierarchical levels in the r-quotient graph. In cases where block labeling has multiple solutions, such as in cycles, use biological information to define as roots (level zero) the blocks that receive extracellular signals.

Another strategy would be to apply agony directly to the interaction graph. However, the computational burden is reduced, and the optimization result is robust by using the r-quotient graph instead. The value of $r$ has to be chosen not too large to avoid one r-block that contains all the species, and not too small to avoid many r-blocks that contain just one species. For the models studied in this paper $r$ equal to one or two is good enough to avoid having just one block or many blocks containing just one species. This is because, even for $r = 1$, the blocks are large enough for these models, so we typically use small values of $r$.

The hierarchical decomposition algorithms used in this paper were implemented in Python (see HiNetDecom in the Availability section).

The application of this procedure to signalling networks with feedback is illustrated in Figs 4 and 5.

**Signal back-propagation, reduced Michaelis-Menten mechanisms, and irreversible reactions.** Although in a signalling cascade the signal usually propagates in only one direction, there are situations when both forward and backward propagation are possible. Similar to the case of feedback, minimal blocks and autonomous pairs can encompass the entire pathway in this instance as well. Signal backpropagation can occur due to enzyme sequestration, a phenomenon in which the active enzyme from an upstream tier of the signalling cascade is sequestered as part of the enzyme-substrate complex [40]. The back-propagation phenomenon disappears under quasi-steady state (QSS) conditions, when the enzyme-substrate complexes have low concentrations [41, 42]. As signalling pathways models often assume QSS, it is useful to have a tool that reduces mass action models by eliminating complexes. The reduced models can then be decomposed hierarchically using methods proposed above. As a result of the reduction, the hierarchical decomposition is improved: some large blocks split into smaller ones. This holds whenever there is sequestration-related back-propagation, whether or not it is accompanied by feedback.

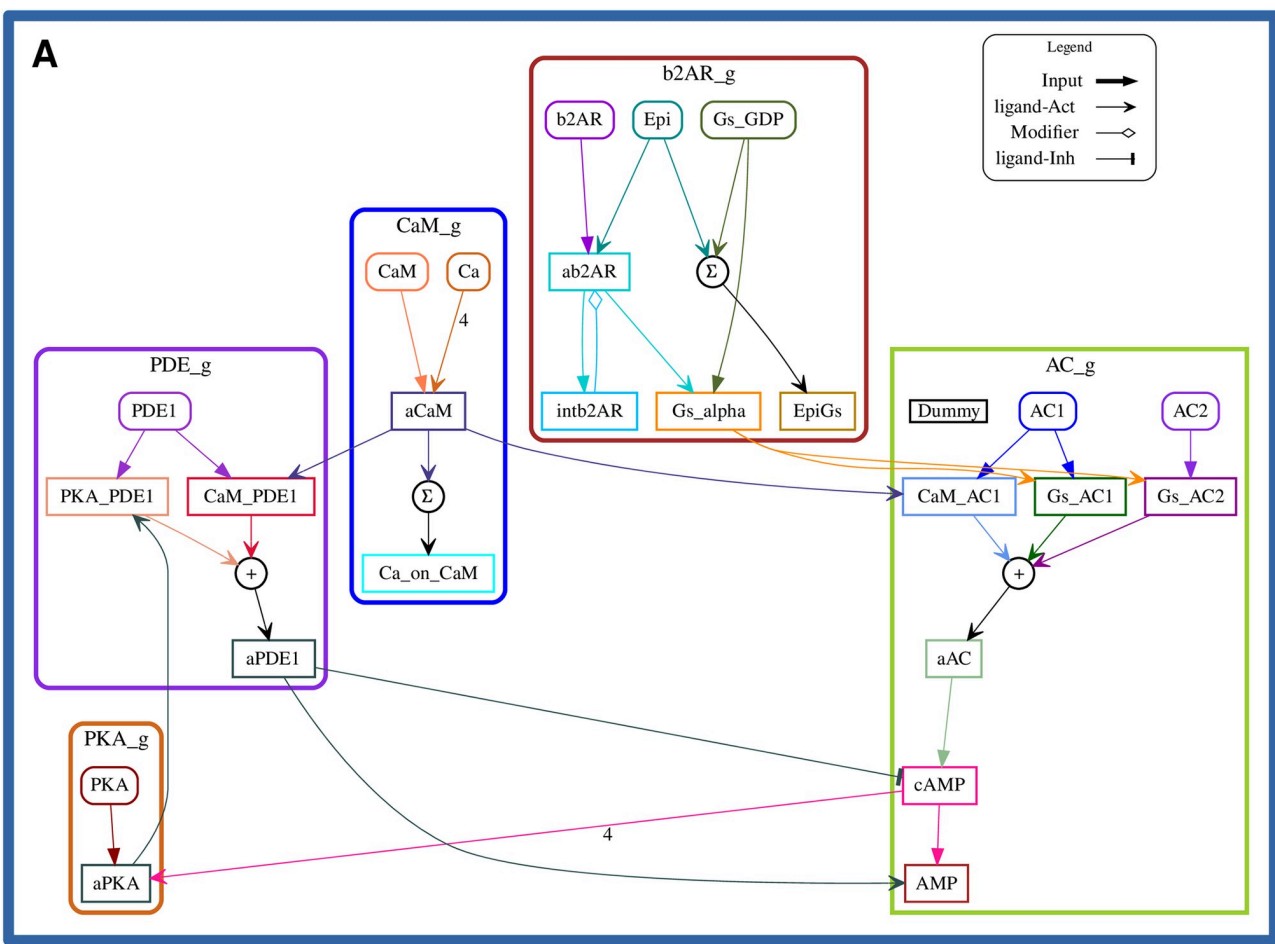

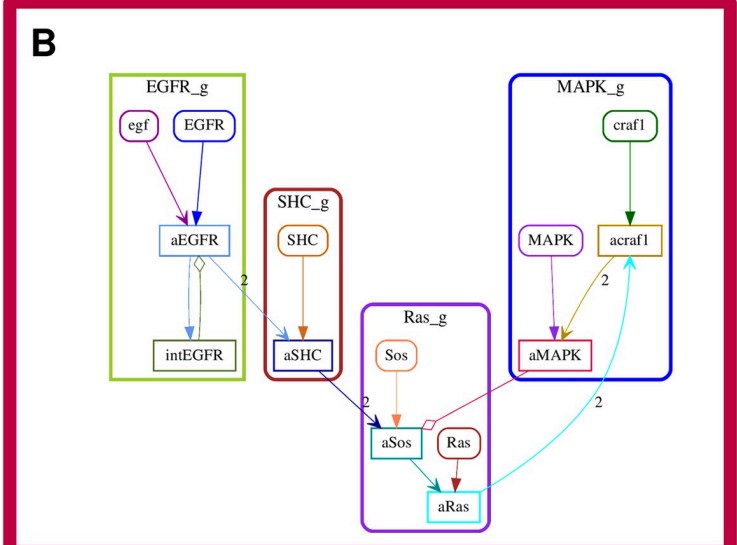

**Fig 3. HillTau versions of models used in current study.** A: Beta-2 adrenergic receptor pathway leading to Protein Kinase A activation (D3 b2AR pathway), implemented in HillTau format. B: Epidermal growth factor receptor pathway leading to Mitogen-Activated Protein Kinase activation (D3 EGFR pathway), implemented in HillTau format.

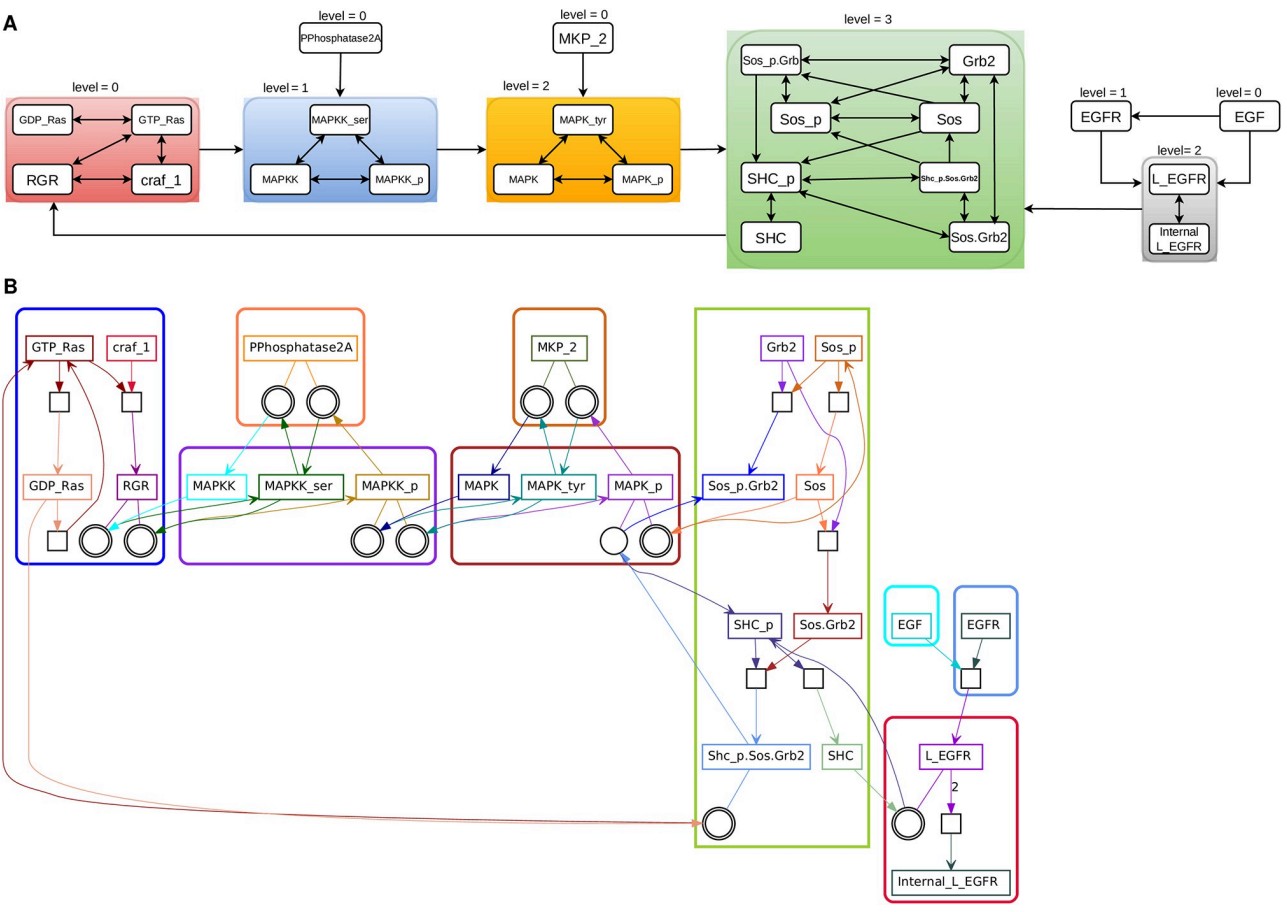

**Fig 4. D4 EGFR pathway implemented in ODE format.** Note that the ODE format implementations are more chemically detailed, but retain overlap with HillTau implementations for several key readouts. Blocks and quotient graph are computed by automated hierarchical decomposition. (A) *r*-blocks and quotient graph after Michaelis-Menten type reduction. One has the same blocks for *r* = 1, 2. (B) *r*-blocks and reaction bipartite graph. For the decomposition, we have considered that the reaction EGFR + EGF ⇌ LEGFR is forward irreversible. The forward irreversibility conditions were verified by numerical simulations.

The QSS reduction of Michaelis-Menten mechanism is based on identifying in the reaction network of motifs of the type:

$$S_i + E_i \underset{k_i^-}{\overset{k_i^+}{\rightleftharpoons}} ES_i \overset{k_{\mathrm{cat}}^i}{\rightarrow} P_i + E_i,$$

and finding all the motifs that share the same enzyme $E_i$.

Let $\mathcal{E}_i$ be the subset of reactions using the same enzyme $E_i$, i.e. $E_j = E_i, \ \forall j \in \mathcal{E}_i$. Then $\forall j \in \mathcal{E}_i$ the Michaelis-Menten mechanism is replaced by a single reaction

$$S_j \overset{V_j}{\rightarrow} P_j,$$

with the rate

$$V_j = k_{\mathrm{cat}}^j \frac{E_i S_j / k_m^j}{1 + \sum_{l \in \mathcal{E}_i} S_l / k_m^l},$$

where $k_m^j = (k_j^- + k_{\mathrm{cat}}^j)/k_j^+$.

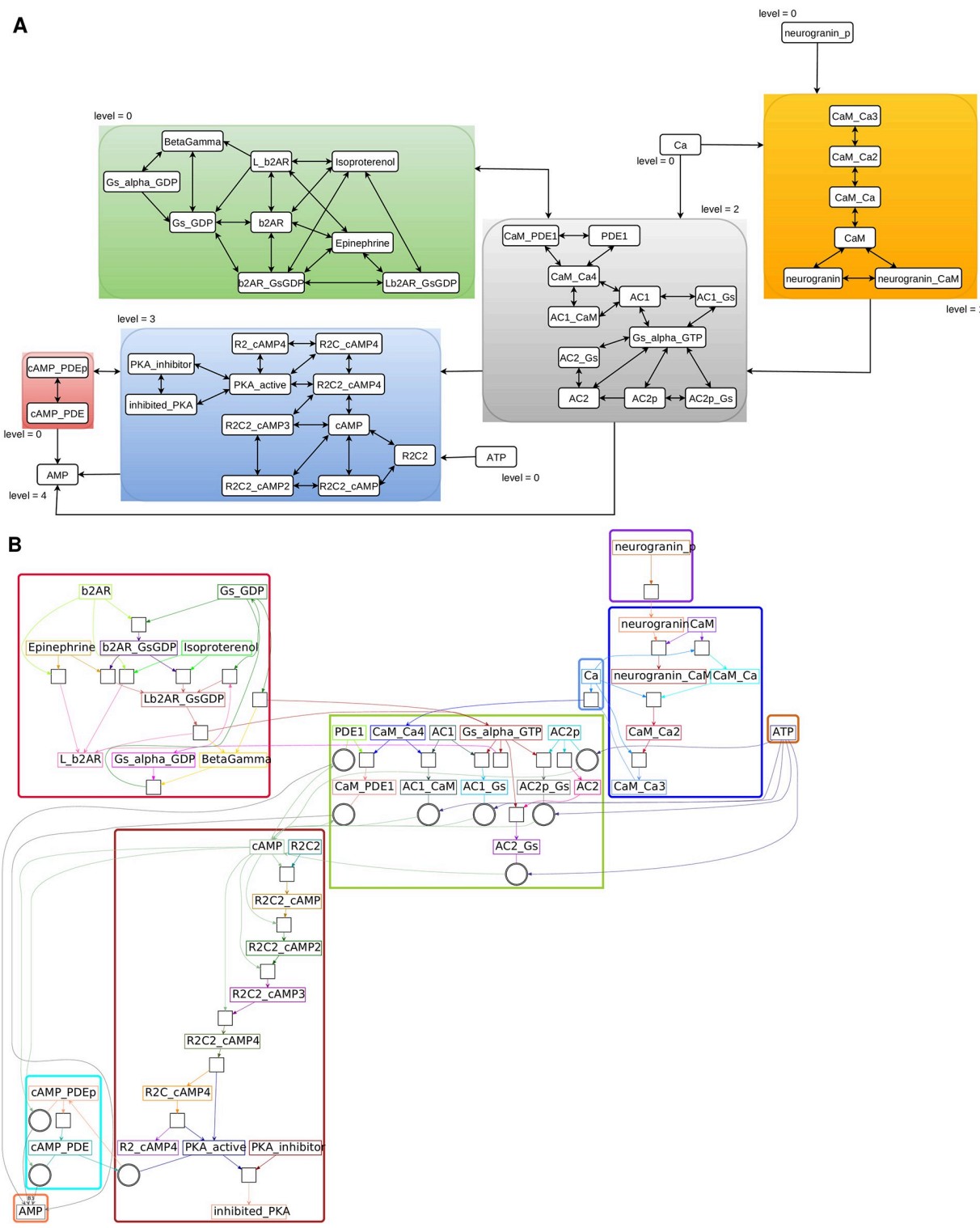

**Fig 5. D4 b2AR pathway implemented in ODE format compatible with SBML.** Blocks and quotient graph are computed by automated hierarchical decomposition. (A) $r$-blocks and quotient graph after Michaelis-Menten type reduction. One has the same blocks for $r = 1, 2$. (B) $r$-blocks and reaction bipartite graph. For the decompositon, we have considered that the reaction CaMCa3 + Ca $\rightleftharpoons$ CaMCa4 is forward irreversible. The forward irreversibility conditions were verified by numerical simulations.

The rates of the reduced reaction depend on the concentration of the substrate $S_j$, but also on enzyme $E_i$ and on the substrates $S_k, k \in \mathcal{E}_i, k \neq j$, that should be added to the list of modifiers of this reaction.

Other sources of signal back-propagation are the reversible reactions connecting species from different levels of the hierarchy. A reversible reaction allows the propagation of the signal in both directions and establishes interaction graph connections in both directions between reactants and products. However, some reversible reaction effectively function in only one direction. We say that a reaction is *forward irreversible* if $R_+ >> R_-$ where $R_+$, $R_-$ are the forward and backward reaction rates. This condition can be verified using numerical simulations or any information about the orders of magnitude of the kinetic constants and concentrations of reactants and products. When it is satisfied we can consider that the reaction is irreversible.

As an illustration, we tested by simulation the forward irreversibility in the signalling model D4 b2AR-PKA. We found several forward irreversible reactions, but the reaction CaMCa3 + Ca $\rightleftharpoons$ CaMCa4 is particularly important for the directionality of the signal propagation. By considering this reaction to be forward irreversible, a large block containing AC1,AC2 and CaM splits into two blocks, one of level one containing CaM and the other of level two containing CaMCa4 (see Fig 5). Indeed, the signal propagates from CaMCa3 to CaMCa4 and not backwards.

Although QSS reduction or forward irreversibility can improve hierarchical decomposition by decreasing the size of some modules, these two approaches are optional in HOSS. r-causality alone, or in combination with agony, can be used to identify sufficiently small modules. Furthermore, QSS reduction and forward irreversibility are applied only during the module definition phase and not during simulation, which implements the full mechanism.

## The HOSS optimization framework

The HOSS software is designed to orchestrate complex, multi-level hierarchical optimizations. To do this it deploys numerous individual optimization steps, each of which fits a subset of a model to a number of individual experiments (Fig 2A). HOSS works on signalling and other models which are subdivided into blocks, typically individual signalling pathways in a signalling network. The blocks are organized into a hierarchy informed by the above mathematical formalism, where each level depends only on signalling input coming from preceding levels, and blocks within a level are independent of each other (Fig 2B). During operation, HOSS reads a configuration file in JSON format, which specifies the metadata and overall optimization parameters, such as optimization algorithm and tolerance (Fig 2B). The configuration file further specifies a weighted set of experimental protocols defined in the FindSim JSON format [26]. Finally, within each block it identifies which parameters are to be adjusted, and optionally their bounds. HOSS calls the FindSim utility [26] to set the parameter vector, and to compute the objective (cost) function giving the accuracy of the model fit for each experiment. The default objective function is the normalized root-mean-square difference between experimental data and simulation readout (2). When several experiments pertaining to different readouts, or datasets of different origins are available for the same model, a consolidated objective function is obtained by combining individual objective functions scaled by weights (3). This consolidated objective function is used in the optimization algorithm which is provided by scipy.minimize. HOSS can employ nested parallelization by simultaneously running FindSim on each experiment within a block, and independently optimizing each block on different processes. For the purposes of subsequent discussion, we refer to the optimization routine (provided by scipy.optimize) as the optimization *algorithm*, and the hierarchical optimization program (provided by HOSS) as the HOSS *method*.

FindSim is the Framework for Integration of Neuronal Data and SIgnalling Models [26]. Briefly, it does three things: 1) reads a model and tweaks its parameters, 2) reads the definition of an experiment and runs it on the model, and 3) compares the output of the model with data from an experiment (Fig 2D–2H). FindSim is agnostic to model definition format and simulator. It currently works with the HillTau [5] format and simulator, and with ODE and mass action models specified in SBML and other formats, and solved using the MOOSE simulator [43]. FindSim utilizes a JSON format file to specify experiment inputs and readouts. Crucially, an experiment defined in FindSim format can be applied to completely different models even using different modelling formalisms, provided the input and output entities are common. We illustrate these capabilities below. In the context of HOSS, we use FindSim on four kinds of experiments applicable to cellular signalling: dose-response, time-series, bar-charts and direct parameter estimates (Fig 2D–2H). FindSim has additional capabilities to handle common electrophysiological experiments [44, 45] but these are not used in the current study.

## Large models overview

For the purposes of this report, we model two signalling pathways in two formalisms each (Figs 3A, 3B, 4B and 5B). The pathways are the beta-adrenergic receptor activation of protein kinase A (the b2AR pathway) and the epidermal growth factor activation of MAPK/ERKII (the EGFR pathway). The reaction topologies of these pathways are based on and simplified from [2]. The two formalisms are HillTau [5], which is an abstracted reduced signalling model specification which maintains direct experimental mapping of selected parameters such as concentrations and rates; and well-mixed chemical kinetics specified in SBML or other compatible formats. In the current study, SBML models are based on ODE dynamics. However, in HillTau, species dynamics are not computed using ODEs, but by a hybrid system.

The ODE models are subsets of the model presented in [2], DOQCS accession 3 https://doqcs.ncbs.res.in/template.php?&y=accessiondetails&an=3. They have been modified slightly as part of the subsetting process. The HillTau models were derived from the ODE models by abstracting the reaction steps. The selected experiments were obtained programmatically from an in-house dataset by selecting for those experiments having stimulus and readout molecules that were present in the models. All models and experiment files used in the optimization are provided on GitHub and were generated to answer specific biological questions in neuroscience independent of the current study.

The composition of the models is reported in Table 1.

## Experimental database

We have used manual curation of the experimental literature to build up a repository of over 350 signalling experiments with a focus on synaptic signalling pathways. There are two key characteristics of this dataset, which drives several of the design choices in HOSS. First, the number of experiments pertaining to each pathway is limited, and even though some

**Table 1. Composition of large test models used in this study.** The number of parameters is contrasted with the number of experiments available to constrain them. The number of data points refers to the count of all data readings in all the experiments, such as successive points in a time-series.

| Pathway | Formalism | Number of species | Number of reactions | Number of parameters | Number of experiments | Number of data points |
|---------|-----------|-------------------|---------------------|----------------------|-----------------------|-----------------------|
| D3 EGFR-MAPK | HillTau | 14 | 7 | 29 | 21 | 97 |
| D3 b2AR-PKA | HillTau | 21 | 12 | 37 | 20 | 129 |
| D4 EGFR-MAPK | ODE | 36 | 22 | 54 | 32 | 241 |
| D4 b2AR-PKA | ODE | 53 | 40 | 93 | 27 | 178 |

experiments provide multiple sample points (Table 1), the constraints remain considerably below the number of parameters even for HillTau models. For instance, many time-series and dose-response curves are asymptotically converging. Hence they may have many points but only constrain a single parameter.

Second, there are frequently overlapping experiments which disagree on the quantitative values of readouts (Fig 6C and 6D). Due to such conflicts within the datasets, a single model cannot be perfectly fitted to data from different laboratories. As a result, some species in some experiments exhibit larger training errors, despite the overall optimization cost being low.

## Results

### Hierarchical optimization outperforms flat optimization for a paradigmatic model with synthetic datasets

In order to illustrate and test the hierarchical optimization method we first use a paradigmatic model of the MAPK signalling cascade, introduced by Huang and Ferrel [46]. The SBML model is available in the Biomodels [47] database. The corresponding ODE system can be found in ODEbase [48] database https://www.odebase.org/detail/1330. The original SBML model consists of mass-action elementary reactions. Because of multiple Michaelis-Menten mechanisms sharing the same enzyme there is back-propagation of the signal and the application of the hierarchical decomposition algorithm to this model results in only one autonomous pair that includes the entire model.

By applying the QSS reduction transformation, the 22 ODEs in the ODEbase model are simplified to 8 differential equations. Notably, 4 species exclusively function as enzymes, and are considered buffered after the transformation (MAPKKK activator, MAPKKK inactivator, MAPKKPase, MAPKPase). As shown in Fig 7A, the reduced MAPK model lends itself to a hierarchical cascade with 3 levels.

We tested hierarchical optimization using time series produced in [49], consisting of 10 *in silico* experiments. Each experiment employed a different concentration of MAPKKK activator. For the flat optimization we used 12 distinct starting points log-uniformly distributed in a hypercube with edges $[10^{-10}, 10]$, and for hierarchical optimization (with parameter scrambling) we used 12 starting points per level. Fig 7B shows that flat optimization takes longer compared to hierarchical optimization in terms of total duration. Additionally, the hierarchical optimization outperforms classical optimization significantly in terms of the objective function value (Fig 7C).

### Black-box, non-gradient optimization methods work well for flat optimization

To scale up our analysis to moderately large models, we utilized the HOSS pipeline on a set of four signalling pathways as described in Table 1. Notably all details required for execution of the optimization pipeline, such as applicable experiments (FindSim files in JSON format), experiment weights, parameter lists, and parameter bounds were incorporated into the HOSS files. Thus a single command triggers execution of a complex pipeline, and a single file orchestrates all the data, models, optimization options, and parameter specification. As a reference, we first ran the HOSS pipeline using flat (non-hierarchical) optimization on the models, employing a number of standard optimization methods in the scipy.minimize library (Fig 8A). Our initial models were initially parameterized manually using inspection of a limited subset of experiments. Following the flat optimization, all of the algorithms produced better fitting models than the start models. This was reflected in the modest improvement in the model-

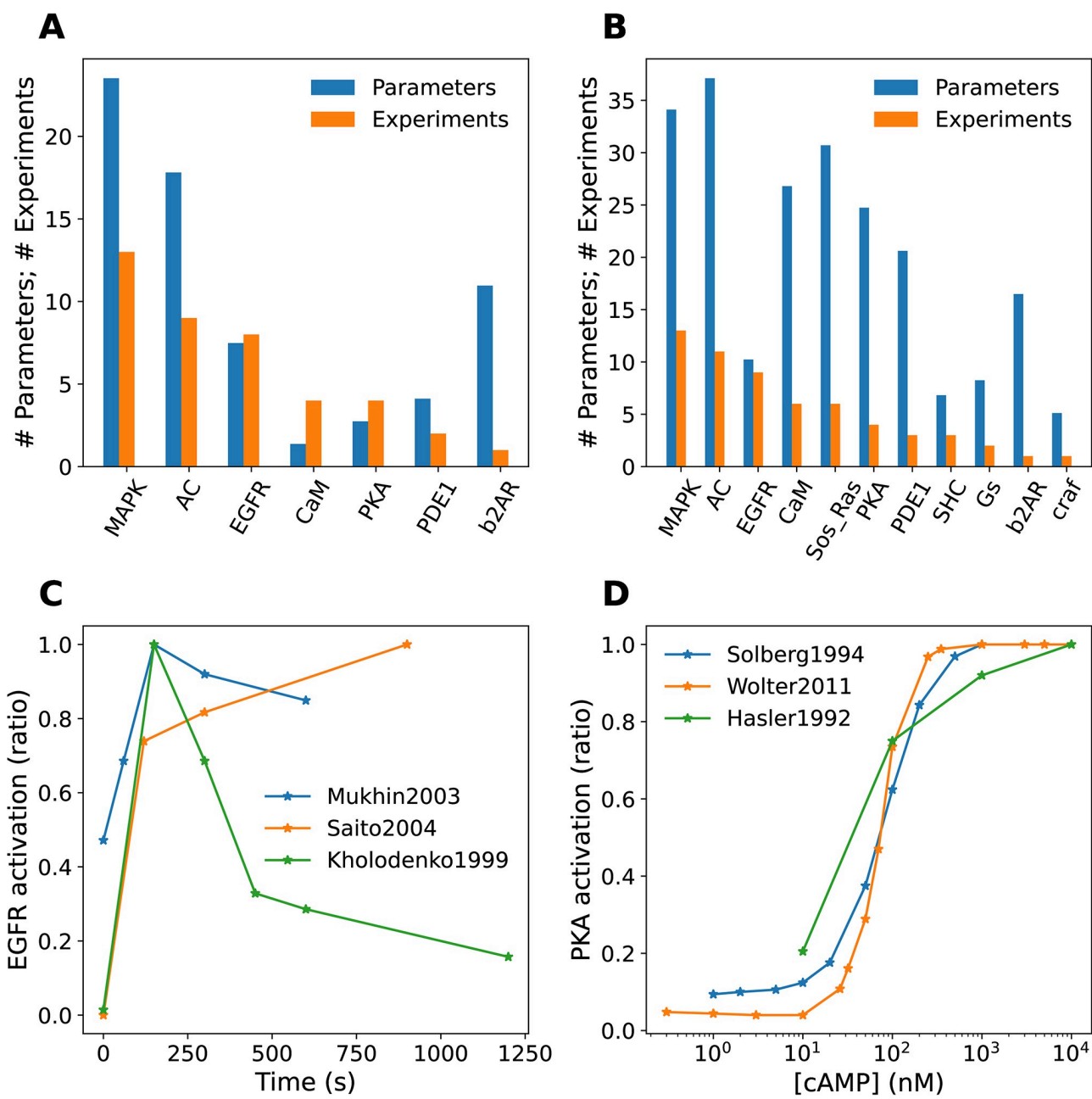

**Fig 6. Features of experimental database.** A, B: Number of parameters (blue) and number of experiments (orange) to constrain them, for different blocks in the model, sorted in order of decreasing number of experiments. In almost all cases the number of experiments falls well short of the number of parameters, that is, the model is underconstrained. A: Reduced (HillTau) models. B: ODE (SBML) models. C, D: Experiments may be inconsistent. C: Three time-series experiments for EGFR activation following a pulse of EGF, normalized to maximal response. These experiments were performed on different cell lines and not surprisingly, the time-courses differ [65–67]. D: Three dose-response experiments for PKA activation by cAMP. These experiments use purified preparations and despite somewhat different conditions the Kd is quite similar [68–70].

fitting objective function, which we refer to as cost (Fig 8B). We found that COBYLA (black-box, non-gradient algorithm based on linear programming and linearization of the problem and constraints) and SLSQP (iterative quadratic programming method, also using linearized constraints) were considerably faster to converge than gradient algorithms such as BFGS

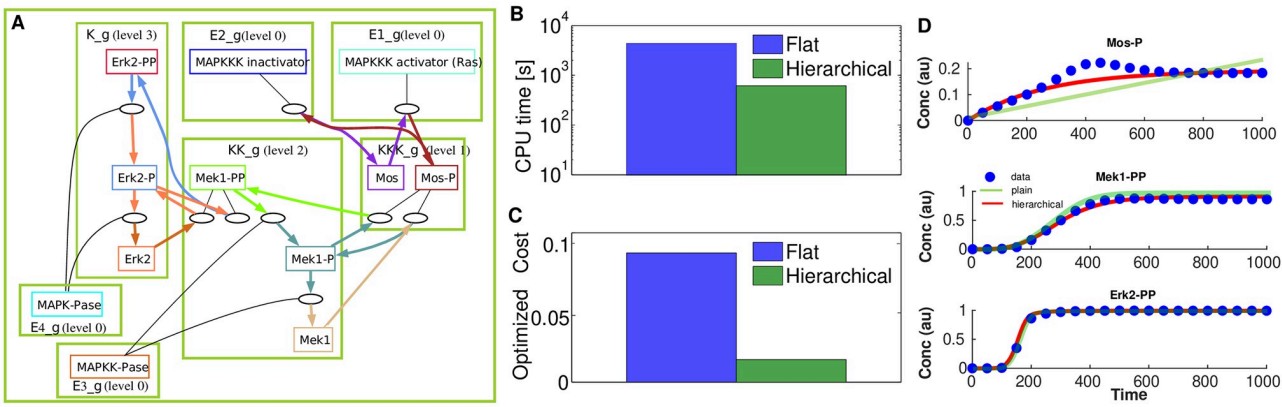

**Fig 7. Optimization of the reduced MAPK model introduced by Huang and Ferrell [46].** This paradigmatic model was optimized using synthetic data. A) Blocks definitions with their levels. Nested hierarchies are made of nested sets of blocks (all blocks of level smaller or equal to a given one) as follows level 0 (E1_g, E2_g, E3_g, E4_g), level 1 (E1_g, E2_g, E3_g, E4_g, KKK_g), level 2 (E1_g, E2_g, E3_g, E4_g, KKK_g, KK_g), level3 (E1_g, E2_g, E3_g, E4_g, KKK_g, KK_g, K_g). B,C,D) Performance of flat and hierarchical optimization.

(quasi-Newton algorithm based on an approximated inverse Hessian matrix) (Fig 8C). COBYLA was more reliable in producing small costs. A possible explanation for this effect is the conflict within the multiple datasets used in the weighted cost (3). This conflict may lead to ill-conditioned Hessians and degenerate quadratic approximations of the cost functions, which disadvantage the BFGS and SLSQP algorithms. Accordingly we used COBYLA for subsequent hierarchical optimization runs.

We have not tested here more precise methods to estimate the gradients, such as forward sensitivity, that may have good performance even with flat cost functions. These will be implemented in future versions of HOSS.

## Hierarchical optimization is more efficient than flat optimization for biochemical models with real datasets

We next tested the HOSS pipeline for hierarchical optimization (Fig 9A). We have shown above that nested hierarchical optimization is more efficient than flat optimization for fitting a medium-sized model with synthetic data. Our results here show that this efficiency carries over to complex real-world cases involving large models (Figs 3, 4 and 5), large but sparse datasets (Fig 6A and 6B), and noisy and sometimes inconsistent data (Fig 6C and 6D). We implemented hierarchical optimization in HOSS as schematized in (Fig 9A).

The signalling reactions from Figs 3, 4 and 5 were subdivided into individual pathways reflecting their biological organization. Within the HOSS configuration file for each model, the pathways were placed in a hierarchy which reflected their position in the signalling cascade (e.g., Figs 2C and 3A). We again tested three different algorithms for optimization: BFGS, COBYLA and SLSQP. We found that hierarchical optimization worked for all algorithms, though COBYLA gave smaller costs than BFGS and SLSQP in most cases (Fig 9B). The runtimes followed the same pattern as for flat optimization, that is, BFGS > COBYLA > SLSQP. We then compared how hierarchical optimization performed compared to flat optimization (Fig 9D and 9E). HOSS gave smaller or comparable costs to flat optimization in all except the ODE-based EGFR model, labeled D4 EGFR. We speculate that a loop unrolling pass would improve the EGFR pathway cost, since there is a feedback loop in the EGFR pathway which violates the hierarchy assumptions. Notably, the runtime for hierarchical optimization was considerably faster in all cases.

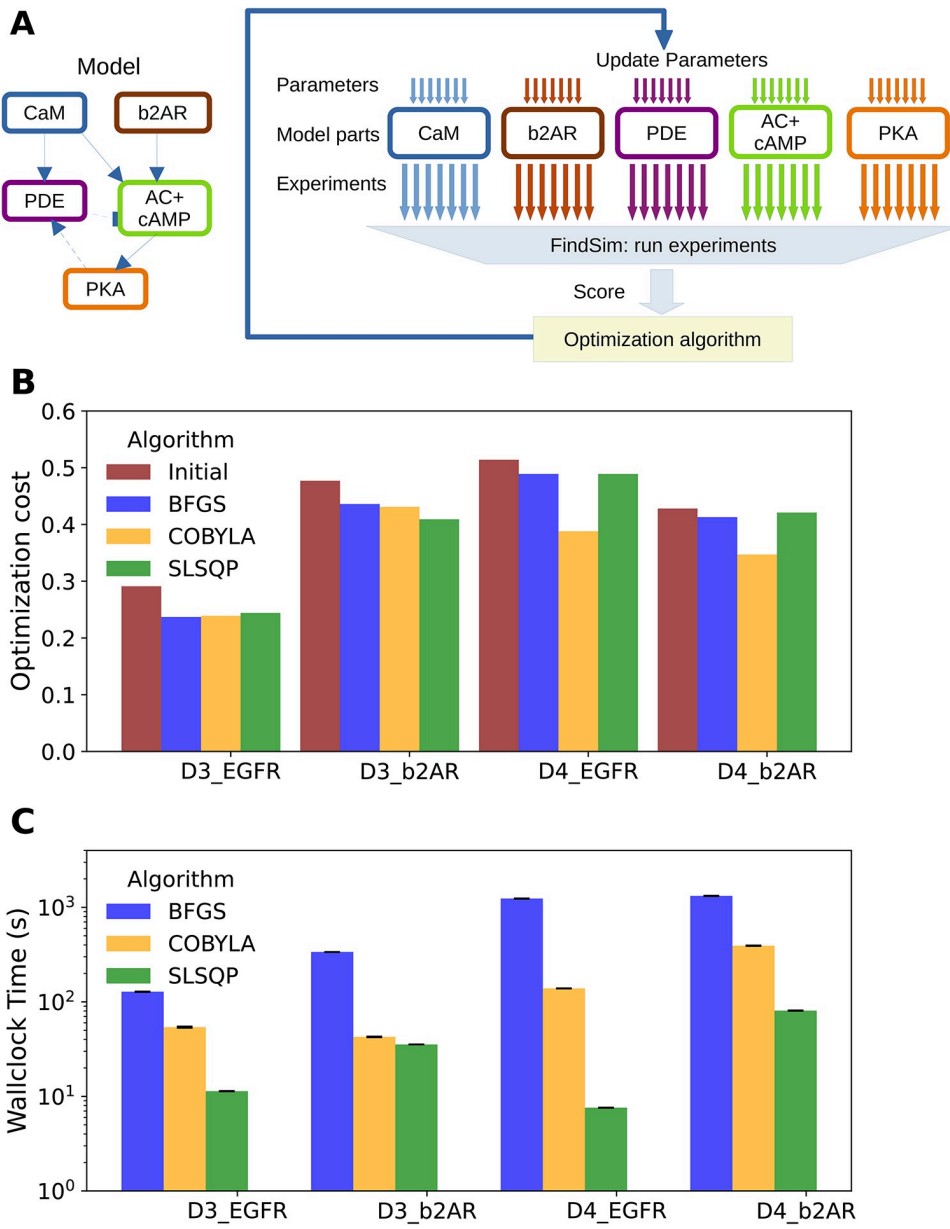

**Fig 8. Flat optimization method.** A: Schematic of flat optimization: Left: Model. Right: Method. All model subsets and all experiments are run in parallel. The resulting costs are combined into a single value used by the optimizer. The optimizer adjusts all parameters across model subsets, for each iteration. B: Barchart of costs for the four different models, comparing the initial cost with the final cost obtained using three different algorithms (BFGS, COBYLA, SLSQP) from the scipy.minimize library. BFGS is a gradient descent algorithm. Note that SLSQP sometimes does not converge to a low cost. C: Barchart of runtimes for the different algorithms. BFGS is always slower. Although SLSQP is typically the fastest algorithm, it sometimes produces high costs as seen in panel B.

## Multistart methods yield lower cost function value: InitScram method

As the basic HOSS algorithm may be susceptible to local minima, we implemented a version which generated a large number of initial models with parameters randomized in a log-normal distribution of half-width scramble Range (scramRange, defined as $b = 1/a = $ scramRange $> 1$ in Eq (5)) (Fig 10A and 10B). This is a known approach, with roots in simulated annealing

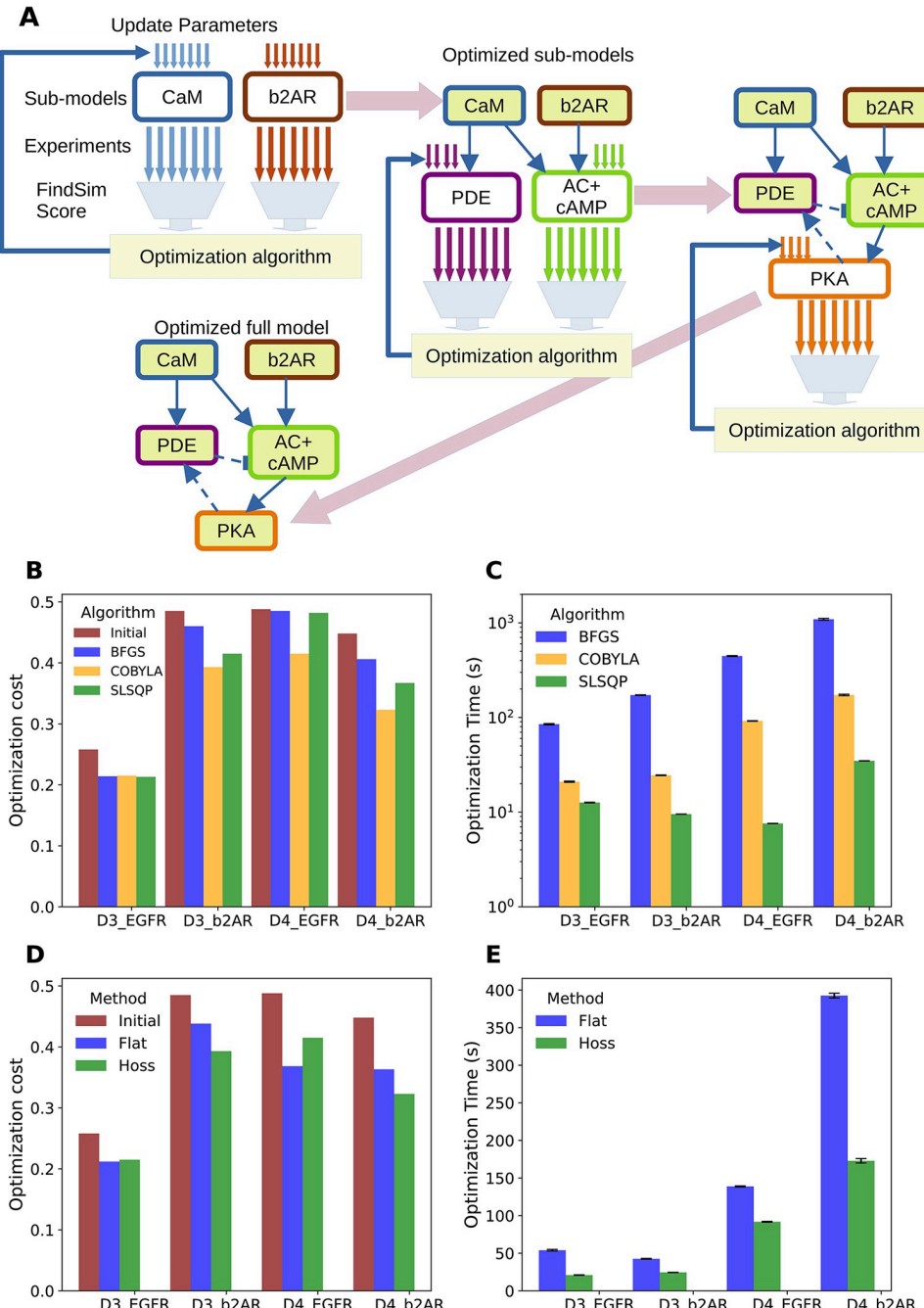

**Fig 9. Hierarchical Optimization on large models.** A: Schematic of hierarchical optimization as implemented in HOSS. First, the upper level of the model hierarchy is optimized, in this case the CaM and b2AR sub-models. Each is individually optimized, and any of the standard algorithms such as BFGS or COBYLA may be employed. Experiments specific to each sub-model are used to compute individual costs and independently update the sub-model parameters. Then, these sub-models are held fixed and the next level of the hierarchy is optimized (PDE and AC+cAMP sub-models). Finally, the lowest level of hierarchy (PKA) is optimized. With this the entire optimization is complete. B: Barchart of costs for the four different models, comparing the initial cost with the final cost obtained using three different algorithms from the scipy.minimize library. C: Barchart of runtimes for the different algorithms. As in the flat method, SLSQP is the fastest. D: Hierarchical optimization vs flat costs using COBYLA. With a single exception, HOSS gives lower or comparable costs. This exception is likely due to relaxation of hierarchy assumptions due to feedback. E: Timing of optimizations run using hierarchical optimization vs flat optimization timing using COBYLA. Hierarchical optimization is faster.

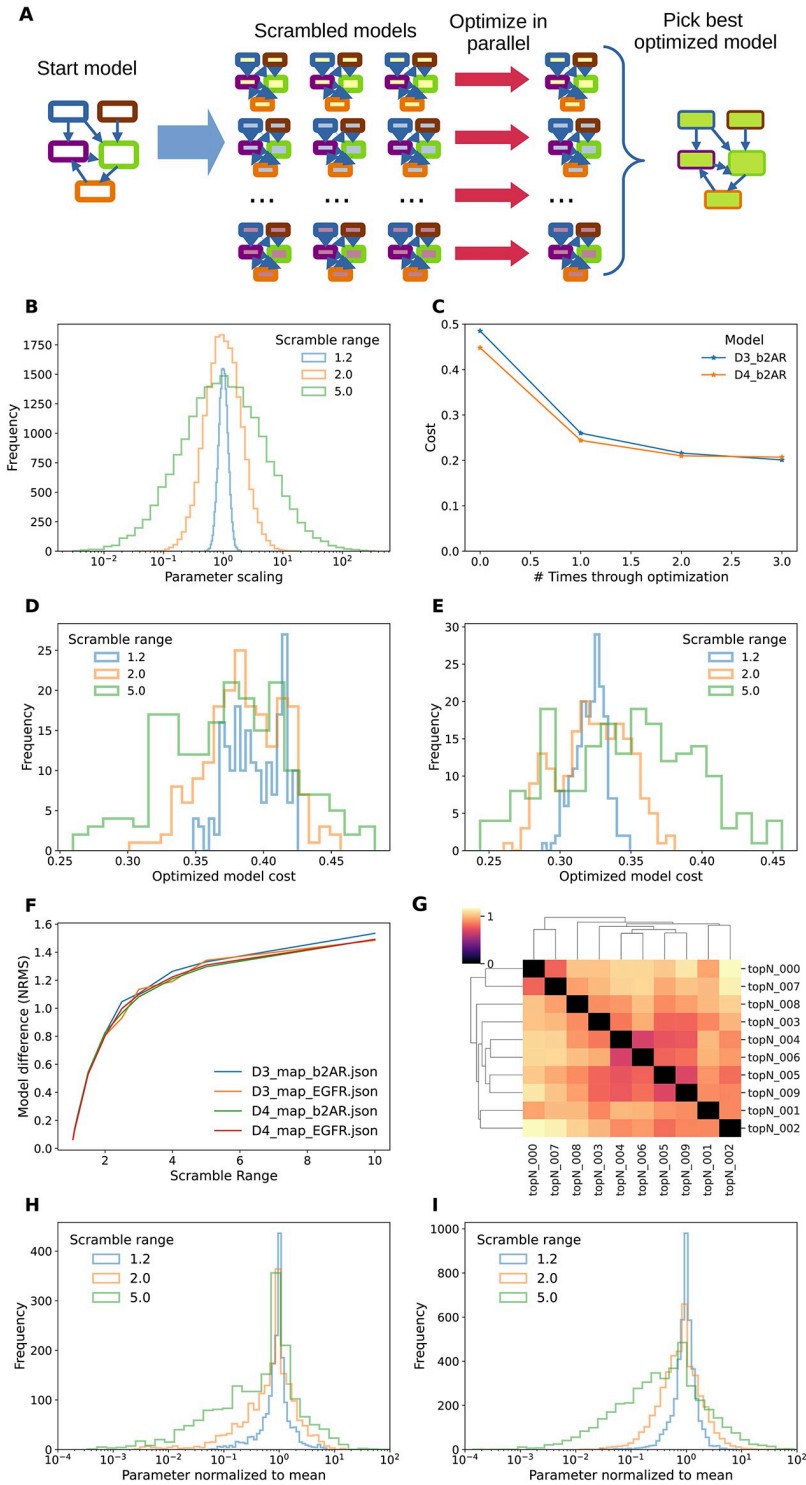

**Fig 10. Multistart (InitScram) method.** A. Schematic of method. B. log-normal distribution of parameter scaling from reference values for scrambleRange (SR) of 1.2, 2.0, and 5.0. C. Improvement of fit over successive optimizations. A second optimization produces a small improvement, and little improvement results from a third round. D. Cost distributions for three values of SR, D3 b2AR model. Note that the peaks are similar but the widths greater for larger SR, hence there are parameter sets with smaller costs (left tail of distribution) for large SR. E. Cost distributions for 3 values of SR, D4 b2AR model. Here the peaks of the cost distribution moves to the left with smaller SR. F. Mapping between parameter scrambling range and NRMS metric for similarity of models shows that this is independent of model. G. Model cost function value cluster-map for top 10 optimized D4 b2AR models. H: Distribution of parameter

scaling for optimized D3 b2AR models, normalized to mean of respective parameter for the best 10 models from that run. The optimized parameters converge very closely to the best 10 means. I: Distribution of parameter scaling for optimized D4 b2AR models. Here the tails of the distributions are somewhat wider, but there is still a narrow peak around 1.0 showing convergence from different start points. Note that peaks are narrower than the initial parameter ranges from panel B.

methods [35, 50, 51]. We extended the HOSS framework to overlay model parameter scrambling and process farming onto the hierarchical optimization method. This is an embarrassingly parallel problem and each of the optimization processes could run in parallel. In the course of these runs we identified one necessary refinement to the algorithm. In some cases, a subset of the initial models took an enormously long time to converge. Thus we implemented a timeout for each elementary minimization run. This may slightly reduce the number of completed runs, but frequently led to considerable improvement in runtime. In an analogy with simulated annealing, we asked if successive rounds of optimization would find still lower minima. We found that multiple rounds of optimization tended to converge rapidly (Fig 10C). Hence in most cases a single optimization step should suffice.

The cost function values resulting from a typical run with 200 initial models fell into a distribution which depended both on model and on scramRange (Fig 10D and 10E). As expected, the width of the cost distribution increased with scramRange. The best fits were at the left of the distribution and in these examples they were obtained with a scramRange of $\sim 5.0$, that is, log-normal random scaling from 1/5 to 5-fold of each initial parameter (Fig 10D and 10E). The costs for these fits were considerably lower than those obtained with plain HOSS. To relate the NRMS divergence between parameters to scrambleRange, we generated a set of models at a series of scrambleRange values, and computed NRMS between each population (Fig 10F). Interestingly, the best few models (lowest costs) were not necessarily very similar in their parameters. We did a normalized RMS comparison of parameters of the top 10 D4 b2AR models and found no obvious clusters (Fig 10G). Using the relationship from (Fig 10F), we observed that the NRMS range of $\sim 1.0$, as seen in these best 10 models, corresponded to a scrambleRange of $\sim 2.0$. This means that the parameters of these models differed by as much as a factor of two. As another measure of the parameter similarity of 'good' models, we plotted the distribution of (model parameter) / (mean parameter) across all parameters taken from the best 25% of models, that is, those whose costs were in the lowest quartile (Fig 10H and 10I). We found that this clustered around one, suggesting that there is indeed a global optimum to which most models converge. Note that this parameter distribution is narrower with a broad tail, as compared to the source model parameter distribution from (Fig 10B). To directly compare the performance of HOSS with the flat method, we ran multi-start optimizations for all four models using the two methods. We generated 200 initial models in each case, and recorded distributions of solution time and of initial and final cost functions (Fig 11). The peak of the HOSS solution time was around one-third that of the flat method (Fig 11A–11D) except for the D3 EGFR model, where there was only a small improvement. Likewise, the cost functions for the HOSS optimization were better than those for flat optimization (Fig 11E–11H), except for D3 EGFR, in which case they overlapped (Fig 11G).

## Multi-stage Monte-Carlo yields further improvements of the cost function value: hossMC method

As a final refinement of our code-base, we implemented a similar model-scrambling step within each stage of the HOSS algorithm (Fig 12A). Thus, each subset of the model was subject to scrambling to give S variants (S $\sim 200$ for a full run). These S variants were individually

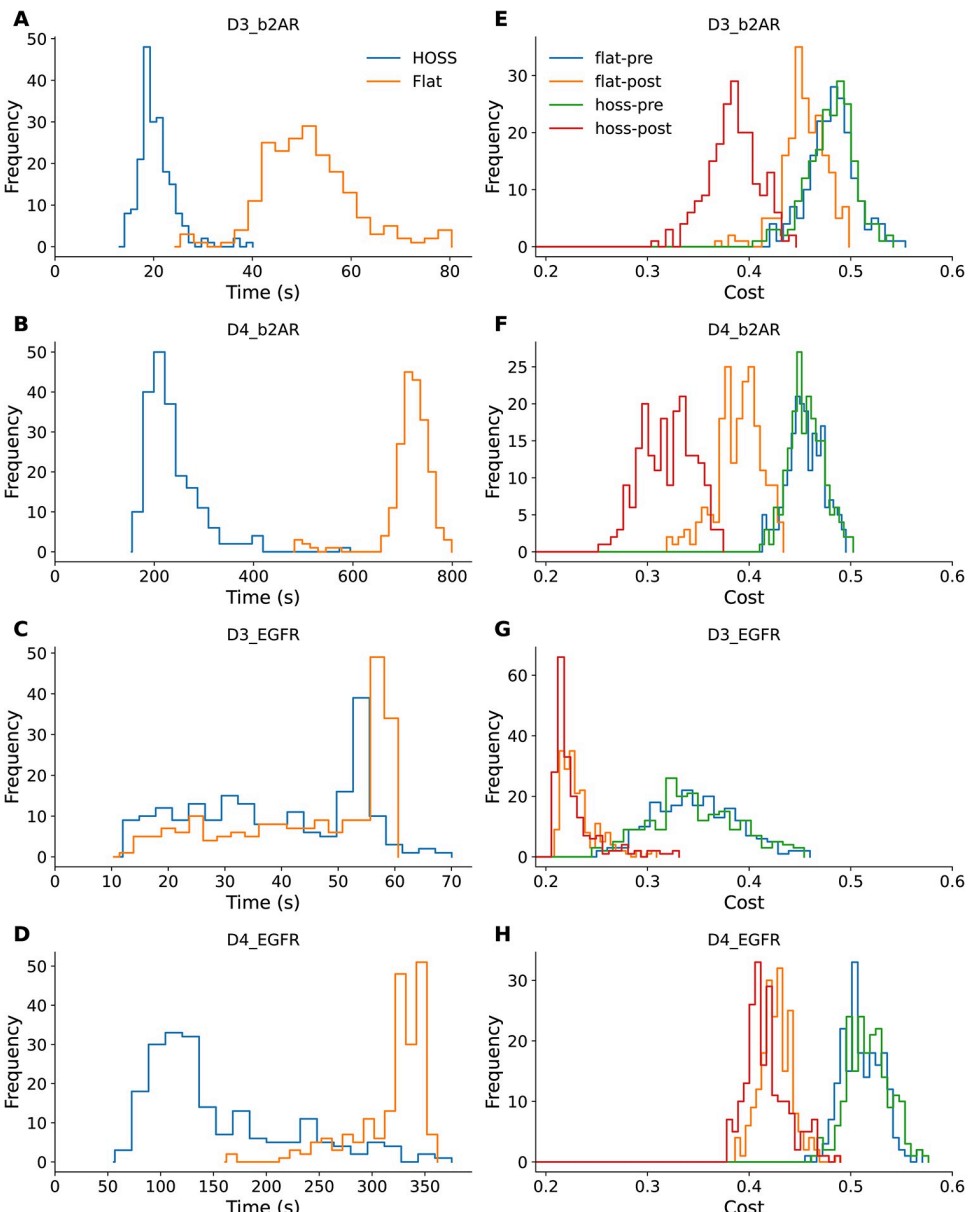

**Fig 11. Comparing HOSS with flat method.** Optimizations were carried out using 200 start points for each method, launched on 16 processes on a 128-core server. Solution times were computed individually for each optimization on each process. A-D: Distribution of solution times for HOSS (blue) and flat (orange) methods, for each of the four models. HOSS converges faster. E-H: Distribution of initial cost (blue, green) and final cost (red = HOSS; orange = flat) for the two methods. HOSS produces better costs in most models but in one case (D3 EGFR) the distribution overlaps with the flat method.

optimized in an elementary minimization step similar to a single stage in the original HOSS method (Fig 9A). If there were multiple model subsets within a given level of the HOSS hierarchy, each was subject to this process to give S optimized variants. The best of each subset were then recombined so as to obtain the top N solutions for a given level. Typical values for N were $\sim 10$. These top N sub-models were then used as separate starting points for further scrambled models for the next level of HOSS, such that we again had S variants to optimize. After the

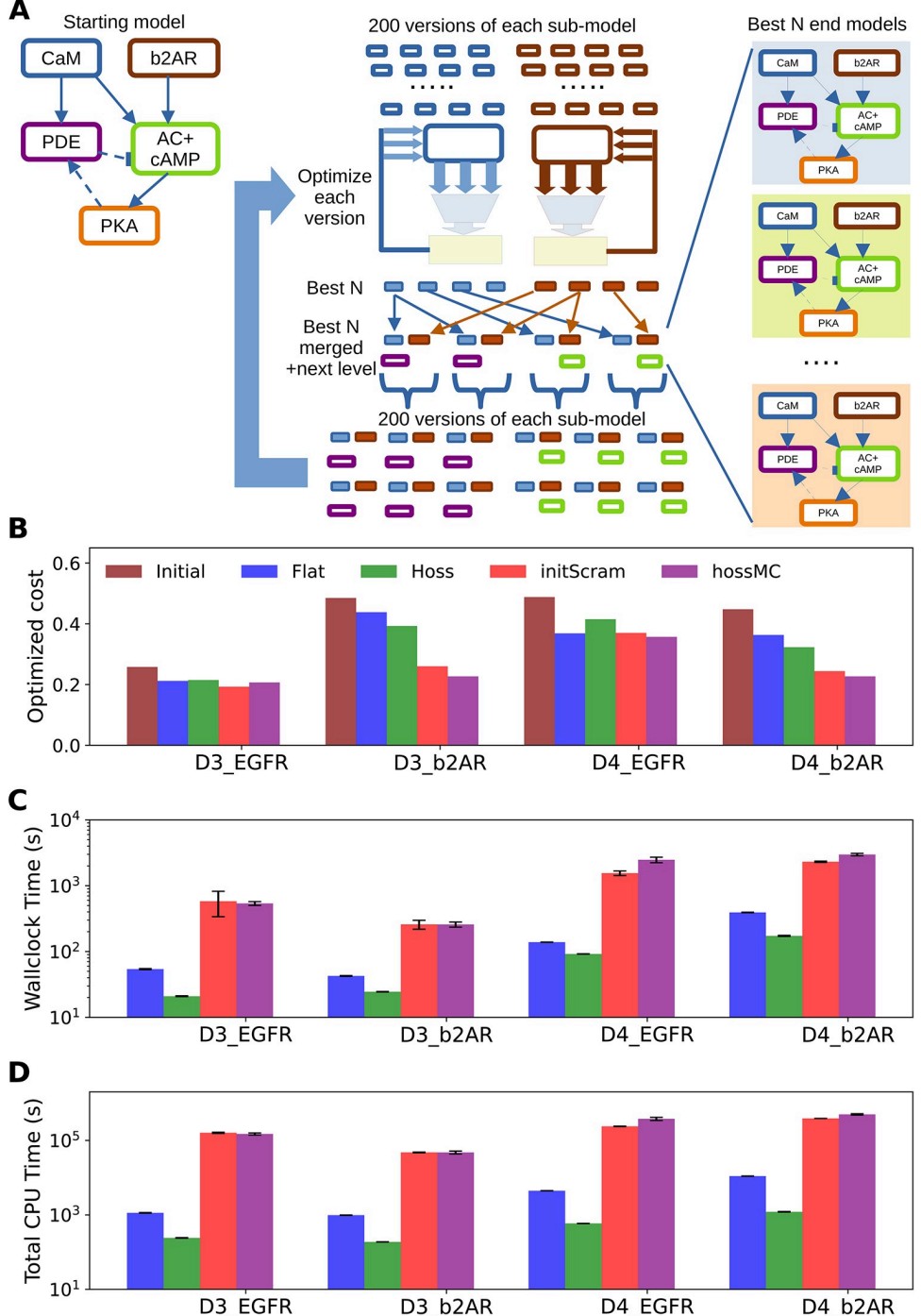

**Fig 12. hossMC method A. Schematic of method.** The model subsets in the first hierarchical level of the model are each scrambled 200 times, and each such starting point is optimized. The best N models (N = 5) are taken from each sub-model and recombined to obtain the overall best N models for the first level. These are then merged with a sub-model from the next level, and these N models are then used as starting points for another round of model scrambling. The 200 scrambled models are again individually optimized as before, and the cycle repeats till we have optimized all levels. The best N merged models are provided as solutions. B, C, D: Comparing the 4 methods (flat, HOSS, initScram and hossMC). B: Cost function values, including the initial cost for reference. The InitScram and hossMC methods worked the best. C. Wallclock time. The plain HOSS method was fastest. The two randomized methods initScram and hossMC were run on 24 processes, but still were much slower because they performed 200 repeats of all optimizations. D. Total CPU time. Here we factor in the number of processes and the parallelization of experiment cost estimation. By this metric, the HOSS method is substantially better than any other, and the two multistart methods are much more computationally costly.

program ran through all levels, we had a set of the best-fitting N models obtained by the overall pipeline. This method generated excellent fits to the data, slightly better than the previous multi-start method initScram (Fig 12B). Wallclock time was similar to that of the initScram method provided there were enough CPU cores available to run all the steps in parallel (Fig 12C). The total CPU time for both randomized methods was also quite similar (Fig 12D).

To summarize the performance of the four methods employed here (flat, hoss, initScram and hossMC), we compared three metrics across the four optimization methods in the HOSS framework. The metrics were the final cost (Fig 12B), wallclock time (Fig 12C), and total CPU time (Fig 12D). As detailed above, the hossMC method was most effective but most CPU-costly, followed closely both in time and model fitting cost by the initScram method. The plain HOSS method was uniformly the fastest, but its cost function values did not compare well with the two multi-start methods (initScram and hossMC) for any of our models. The conventional flat method is not a good choice by any criterion.

## Discussion and conclusion

We have developed a pipeline for hierarchically optimizing large signalling models with hundreds of parameters. We show that hierarchical optimization gives better model fits, and does so faster than conventional flat optimization. We extend this approach to two further methods which use Monte Carlo sampling of multiple parameter start points to give still better final models.

### Model provenance and modelling disease variants

Complex biological models, and signalling models in particular, frequently draw upon diverse sources of data. Such models are often hand-tuned, and such tuning may be very effective because it draws upon expert intuition and implicit knowledge about the behaviour of familiar pathways. However, many model parameters are adopted from the literature without clearly documenting the parameter optimization procedures or the data used in these procedures. This makes model provenance problematic. How did the modeller end up with a particular set of parameters? The HOSS framework introduces model optimization pipelines that are efficient, scalable, repeatable and above all, transparent. The development of a well-structured optimization configuration format in HOSS ensures that all experimental data and model choices, their weights, and all hyperparameter selections are as clearly defined as the algorithms and the simulators. This emphasis on provenance is designed to place the HOSS framework in line with FAIR principles [52]. We highlight two use cases to illustrate how HOSS supports reuse. First, model rederivation: A different scientist may feel that some of the original experiments should be considered more authoritative than others. This can be done simply by assigning a greater numerical weight to the selected experiments, rerunning the pipeline, and seeing what changes in the resultant optimized model. Similarly, a researcher could include some new experiments into the dataset against which the model is to be optimized. This simplicity of model derivation brings a more data-driven flavor to debates over model assumptions and how well they represent the known experimental literature. As HOSS is agnostic to model formalism, it follows that these comparisons could even extend over distinct models implemented with different formalisms (e.g., HillTau vs mass action chemistry). Although not yet implemented, the same principles may apply to optimizing qualitative models such as Boolean networks.

Second, The HOSS structure is highly effective for model specialization. A researcher may wish to make a family of models for different disease mutations, based on a dataset of readouts for experiments in a set of mutant animal or cell lines. Using the HOSS pipeline, it is

straightforward to replace the original (wild-type) experiments with the respective mutant line experiments, rerun the optimization, and obtain disease-specific models. Thus the HOSS framework encourages best practice in developing complex models which can be easily reused.

## Large models and large datasets

HOSS is scalable. This is in large part due to the efficiency of the hierarchical optimization core method we have described. Based on this, we have shown that even large models can be optimized quickly. Beyond this, HOSS organizes systems optimization problems in a modular manner which scales well with complex models and datasets. As a key part of this, HOSS organizes models into hierarchies, within which data, parameter choices, and multiple optimization stages of a pipeline can be triggered using a single command. Thus, once it is set up, a HOSS optimization run does not require many steps of inspection and tweaking by the investigator, and is simple to incrementally extend with new experiments and updated models. Rerunning a pipeline is trivial, and is limited only by computational resources. Several tools also provide model optimization (e.g., COPASI [50]).

Model building is not limited just by resources and datasets, but also by how manageable is the organization of the dataset. The traditional way to associate model parameters with experiments is to provide citations (e.g., refs: DOQCS [53], BioModels [47], ODEbase [48]). This is neither complete, due to the previously mentioned lack of documentation, nor automated, because every iteration of the model would, in principle, require human intervention to produce or find new data, reorganize it and reparametrize the models. Several efforts have sought to reorganize experimental data into a standardized machine-readable format [54, 55], and HOSS uses the FindSim format to do so [26]. The organization of a HOSS pipeline lends itself to version control, since every component of the pipeline is a file in a standard location and standard format. Specifically, the HOSS configuration file is in JSON, the model definition files may be SBML or HillTau, and the experiment specification files are FindSim JSON files.

HOSS encourages the clear subdivision of models and experiments into groupings around individual signalling steps, such as the activation of a kinase by its immediate second messengers. This has implications for experimental design geared to tightly defining large signalling systems, because it lays out the kinds of experiments that are needed to achieve full coverage of all the reaction steps. Notably, we find that two kinds of experiments can greatly tighten parameters: local experiments that probe input-output properties of a given signalling step, and readouts of all key intermediates along a receptor-driven pathway to ensure that signal propagation remains intact.

## Model degeneracy, granularity, and completeness

In cellular signalling models, it is now clear that many parameter combinations may yield the same input-output properties. The origins of this *degeneracy* could be epistemic and due to data sparseness [56], but also biological; being a feature of cells themselves, that can perform the same biological function in multiple ways [57–59]. The HOSS framework may provide a useful tool to study such degeneracy. Most directly, the two Monte Carlo methods supported by HOSS (initScram and hossMC) generate multiple 'good' models, which can be tested for degeneracy (e.g., Fig 10). Because HOSS is agnostic to model detail, simulator, and model formalism, it also lends itself to asking how model granularity affects degeneracy. We have previously suggested that it is useful to develop a family of models at different resolution for any given signalling system [42, 60, 61]. HOSS is well equipped to facilitate this, as it can use the same experimental dataset for models at different detail. We demonstrate this in the model choices in this paper, since the D3 models using HillTau, and the D4 models using the

MOOSE simulator [43], are parameterized using overlapping sets of experiments, separated only by the fact that some experiments in the D4 set depend on molecules that are not defined in the simpler D3 models. Model completeness, referring to how well a model incorporates all necessary details to accurately representing a system or phenomenon, is quite difficult to ascertain in biology as it is in all scientific fields confronting theory and experiments [62]. Several methods have attempted to explore model topology space along with parameters [8, 9, 63, 64], but HOSS supports a more pragmatic interpretation: Is a model complete enough to account for a given set of observations? It does so by trying a large number of possible parameter sets and seeing whether any of these initial conditions result in well-fitting models. A failure to do so suggests that the model topology may need to be reconsidered. We have previously illustrated the behaviour of a series of models of activity-driven synaptic signalling at different levels of granularity, and show that more detailed models fit additional features of the response [5]. However, an overly detailed model can lead to over-fitting if the data is not sufficiently rich. We suggest that multi-grain hierarchical approaches, including automated model granularity (level of detail) selection, may represent a future evolution of hierarchical optimization.

## Acknowledgments

We thank Pawan Kumar for help with supervision of AT, and Clement Raspail for providing a new, improved version of the Michaelis-Menten reduction code.

## Author Contributions

**Conceptualization:** Ovidiu Radulescu, Upinder S. Bhalla.

**Data curation:** Nisha Ann Viswan, Upinder S. Bhalla.

**Formal analysis:** Alexandre Tribut, Manvel Gasparyan, Ovidiu Radulescu.

**Funding acquisition:** Ovidiu Radulescu, Upinder S. Bhalla.

**Investigation:** Nisha Ann Viswan, Alexandre Tribut, Manvel Gasparyan, Ovidiu Radulescu, Upinder S. Bhalla.

**Methodology:** Ovidiu Radulescu, Upinder S. Bhalla.

**Project administration:** Ovidiu Radulescu, Upinder S. Bhalla.

**Resources:** Ovidiu Radulescu, Upinder S. Bhalla.

**Software:** Alexandre Tribut, Manvel Gasparyan, Upinder S. Bhalla.

**Supervision:** Ovidiu Radulescu, Upinder S. Bhalla.

**Visualization:** Upinder S. Bhalla.

**Writing – original draft:** Ovidiu Radulescu, Upinder S. Bhalla.

**Writing – review & editing:** Ovidiu Radulescu, Upinder S. Bhalla.

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
