## [Decision Letter · Decision Letter 0]

28 Aug 2024

Dear Prof. Bhalla,

Thank you very much for submitting your manuscript "Hierarchical optimization of biochemical networks" for consideration at PLOS Computational Biology.

As with all papers reviewed by the journal, your manuscript was reviewed by members of the editorial board and by several independent reviewers. In light of the reviews (below this email), we would like to invite the resubmission of a significantly-revised version that takes into account the reviewers' comments.

We cannot make any decision about publication until we have seen the revised manuscript and your response to the reviewers' comments. Your revised manuscript is also likely to be sent to reviewers for further evaluation.

Sincerely,

Anders Wallqvist

Academic Editor

PLOS Computational Biology

Marc Birtwistle

Section Editor

PLOS Computational Biology

Reviewer's Responses to Questions

**Comments to the Authors:**

Reviewer #1: In the manuscript "Hierarchical optimization of biochemical networks", Viswan et al. describe an approach for the hierarchical decomposition of estimation problem. Instead of solving the full optimisation problem directly, the authors propose to construct a sequence of optimisation problems based on the topology of the biochemical process.

To tackle increasingly large models of biological processes, parameter estimation methods have to be efficient and scalable. To achieve this, a variety a novel concepts have been proposed, including hierarchical optimisation strategies for observation and noise parameters. Here, the authors propose to nested hierarchical optimization to improve parameter estimation further. Unfortunately, the propose is not compare to state of the art method and a discussion of related approach is missing. In my opinion, the extended case with feedback appears similar to the dependent input approach by van Riel and Sontag (DOI: 10.1049/ip-syb:20050076). Furthermore, the paper would profit from an assessment of limitation and applicability, and a comparison further comparison with state-of-the-art methods (see comments below).

Major:

1) The manuscript should discuss related approaches, e.g. the method by Kotte & Heinemann (https://doi.org/10.1093/bioinformatics/btp004) and van Riel & Sontag (DOI: 10.1049/ip-syb:20050076) which also uses a decomposition of the overall problems. As these methods have not been widely used, it would be important to discuss why the authors believe that this will be different for their approach.

2) As there are already hierarchical approaches published for ODE models of biochemical processes (e.g. reference 25 which you include), the author should in my opinion choose a much more specific title. Furthermore, the reference 25 and its extensions in several directions (see for a summary: https://mediatum.ub.tum.de/doc/1625172/a5r8m48zdiyp4qhwmhuqrylg2.PhDThesis_YannikSchaelte_20220725.pdf) should in my opinion be mentioned in the introduction.

3) L. 18-21: "In particular, parameter optimization methods have been implemented in a fragmented manner [9–17]. This is in part due to the very wide diversity of experimental inputs used to constrain such models, but also due to the inherent contradictions and incompleteness of the parameter constraints." - It is unclear for me in which regarding the authors considered the implementation of optimisation methods to be fragmented. There are various comprehensive optimisation packages with a broad support, and even in systems biology there are toolboxes such as COPASI, D2D and PyPESTO supporting dozens of optimisers which allow for constraints.

4) L. 87-88: "In the HOSS calculations we perform two levels of scoring. First, for each experiment for which the sub-model is tested" - At this point submodes have not ben formally introduced. I would recommend to change the order and to start with a comprehensive model formulation and a definition of submodes, before specifying the objective function.

5) L. 87-98: In my opinion the use of statistical interpretable objective functions is highly beneficial. For the objective function used by the authors for an individual dataset (Eq. 2), corresponds to the case with additive normally distributed measurement noise with mean zero and standard deviation equal to the maximum of the observed output. This on its own might be questions, but the interpretation of the sum become even more problematic (Eq. 3).

It would be interesting to know why the authors did not chose an objective functions which allows for statistical meaningful uncertainty quantification.

6) L. 103-105: "we use multistart optimization, by launching local search procedures from randomly chosen starting points generated uniformly in logarithmic scale" - It has been shown that using log-transformed parameters for optimization (not only initial sampling) improve performance tremendously (https://doi.org/10.1093/bioinformatics/btz020 and your reference [26]). It would be interesting to see what can be gained here.

7) Eq. 6: The readability of Eq. 6 could in my opinion be improved. I would recommend to stick to standard terminology and to defined the optimal points directly, e.g. p_{K-1}^* = \\arg \\min_{p_{K-1}} {...}, and to use "subject to" to refer to the conditions in the next level.

8) L. 154-156: "1) if a species is in the subset I, then all the reactions consuming or producing this species are in the corresponding reaction subset J, namely if i \\in I then j \\in J whenever S_{ij} \\neq 0.$ - The statement "if i \\in I then j \\in J whenever S_{ij} \\neq 0" implies in my opinion that all species in I are influences by all reactions in J. This is not what the first part of the sentence states and seems to be awfully restrictive. Please check.

9) Conceptually, the nested decomposition results smaller subproblems. Yet, the issue is that the inference on the small module does only consider the data which can be directly mapped to the module. Consider the reaction network R1: A -> B with rate k1*A, R2: B -> C with rate k2*B, and A and B being measured. The method would infer a small module mit A and R1 only from data in A, but clearly measurements of B provide additional information, in particular if they have a higher resolution.

It would be interesting to assess how much information os lost.

10) The formulation of the nested decomposition depends on the existence of a directionality in the graph (as shown in Figure 1). I assume that many models cannot be decomposed. Did the authors show for which fraction of models a nested decomposition is possible? The could be easily done for the Biomodels database and would -- if the fraction is high and the models are substantial is nice -- increase the impact of the contribution. Even better would be the evaluation of the PEtab benchmark collection taking also the data mapping into account.

11) L. 333-334: "For the purposes of this report, we model two signalling pathways in two formalisms each (Figure 4 A-D)." - I would have preferred if already published models were used. The formulation of own models and datasets raises the questions if they are tailor-made for the proposed approach.

12) L. 348-349: "the number of experiments pertaining to each pathway is limited, and

considerably below the number of parameters" - In my intuition this should be a very problematic setup for the proposed method. If a submodel is non-identifiable based on the corresponding data, parameters might be chosen which do not allow to achieve good estimates in subsequent iteration. I would appreciate if the authors could comment on this to complement the section of degeneracy.

13) L. 386-387: "As a reference, we first ran the HOSS pipeline using flat (non-hierarchical) optimization on the models, employing a number of standard optimization methods in the scipy.minimize library (Figure 7 A)." - For a meaningful comparison, state of the art methods which are tailored to the problem class should ideally be used. For ODE models, these approaches are for instance implemented in D2D, PEtab.jl and pyPESTO. In particular, for gradient based optimisation, sensitivity equation based formulation should be used.

Minor:

L. 21-28: I see the problem mentioned here, but I do not see how this supports / ore is even connected to the previous statements in the paragraph or the contributions of the paper in general.

Indeed, the authors seem to fall in the same trap (see line 345-348).

L. 32-34: "The current paper focuses on standardizing the calibration and optimization stages of model development, given a large but incomplete set of experimental data." - In my opinion there is nothing like a "complete dataset". I guess the authors want to refer here to having only partial observation, but this should be clarified. Furthermore, it would be interesting to know how the authors see the work being related to the establishment of the PEtab format (https://doi.org/10.1371/journal.pcbi.1008646).

L. 60-61: "We illustrate its use on an extant database of over 100 experiment definitions" - The term "experiment definitions" is not well define and should be clarified to avoid the impression that the dataset is much larger than for other models.

L. 66: "parametric optimization problems" -> "parameter optimization problems"

L. 101: "S is a space of constraints" - S does not seem to describe the constraints but the set of admissible parameter vectors. Furthermore, "S" is used for multipel different thing. I would suggest to chose here a different symbol to avoid confusion.

L. 111-112: "We refer to this procedure as parameter scrambling. Despite its simplicity, multistart optimization with logarithmic sampling has proven to be effective in benchmarks of biochemical pathways [26]." - The reference appears inappropriate as [26] optimises xi = log(p) (using p = exp(xi) in the model) and does not only sample the starting points differently.

L. 150-151: "Some species forming a subset BS ⊂ S are buffered, and their concentrations are kept constant." - This statement is not clear for me. Are the authors referring to conserved quantitates? If so, "buffering" might not be the right term.

L. 172-174: "A pair of species (i, j) ∈ E defines an edge from i to j if and only if there is a reaction that consumes or produces the species j, and its rate depends on the concentration of the species i." - Please mention here or in the sentence before that you are considering direct graphs?

L. 378: "Black-box, non-gradient optimization methods work well for flat optimization" - I'm puzzled by this observation. In our experience in particular for problems with relative flat objective functions, gradient-based methods are fare superior. Yet, this only holds of gradients are evaluated accurately, e.g. using forward sensitivity analysis. The dependence of results on such choices could be discussed.

L. 426: "Multistart methods yield lower optimization cost: initScram method" - I recommend to avoid the term "optimization costs". I would used standard terms such as "objective function value" to avoid the confusion with the "computational cost".

Reviewer #2: This is a well written paper addressing an important problem - optimization of signaling pathway models. The introduction nicely reviews the current state of the art, and explains why the problem is critical.

My main comments relate to definition of the hierarchies, autonomous pairs and SCC. Specifically:

Lines 172-173: sounds as if i is causal to j, but then in lines 174-175 you define causality with an example of j causal to i. I think it would be less confusing to use the same causality in both sentences. From the definition of SCC on lines 178-179, I would label A, D and E in Fig 1 as SCC. However, this definition is different than you give in Figure 1. In caption to Figure 1, it seems to me that the definition of SCC doesn't apply to b, which is only connected to c and f, but not a, d or e. So, either you need to change this sentence or remove b from being a scc. Possibly this applies to some other components.

Also, can you label autonomous pairs in this graph? The definition of J1 (line 191) is confusing. How can it include reactions _producing_ species from I1 if I2 has no incoming connections?

Fig 2 and Fig 6: I thought I understood how levels were defined, but these figures calls that into question. How is the pink block with RGR, Ras_GDP of rank 0 since it has an input. In Fig 6, how can a block be of multiple levels? I thought level 1 meant no inputs, which means it cannot be level 2. This needs to be clarified. Also, what is the difference between rank and level? They seem the same, in which case only one of those terms should be used.

Minor comments:

In describing types of models in the introduction, the authors leave out stochastic implementations of biochemical reactions. I'm wondering why these are excluded.

Line 243 - explaining agony: If r has to be not too small to avoid r-blocks with 1 species, why do the authors use models with r equal to one?

Line 266: There appears to be a verb missing in this clause

Line 348-349: In quantifying number of experiments and demonstrating parameters are under-constrained, does that take into account that some experiments measure multiple time points? It should, since comparing multiple time points doesn't increase number of parameters, though I still expect the parameters to be under-constrained.

FIg 4: Text in this figure needs to be much bigger.

Reviewer #3: Summary:

The manuscript “Hierarchical optimization of biochemical networks” by Viswan et al. introduces HOSS, a method to break down large system models into individual pathway blocks organized in a nested hierarchy, allowing for efficient parameter optimization at each level. However, there are several key issues that limit the overall usefulness of the work. One of the key issues is that it is unclear how the hierarchical decompositions handle a signaling network with significant feedback mechanism. In my opinions, feedback mechanisms are almost always present in biological signaling pathways and play crucial roles. If HOSS can only effectively break down system without feedback, it significantly diminishes the utility of the method.

Questions:

In line 210, authors wrote, “The autonomy of these resulting blocks is only approximate, but it allows us to benefit from hierarchical optimization.” What exactly does HOSS do to approximate signaling pathways with feedback? Does HOSS ignore the feedback mechanisms, break down the signaling pathways, and optimize the parameters in each block independently, then reintroduce the feedbacks to estimate their parameters, fix those feedback parameters, and then re-optimize the parameters in each block, iterating this process?

What are the specific benefits of hierarchical optimization when feedback mechanisms are present? Apart from reducing Wallclock time, which is not a major concern for me since estimating hundreds of parameters typically takes a few hours at most, what other advantages does this approach provide?

For example, in Fig 1., if there are two feedbacks loops from F3 to B0 and A0, can HOSS still break down the signaling diagram? How would HOSS approach optimizing the parameters in this scenario?

If the feedback in the diagram is critical to the system and my goal is to estimate the parameters within the feedback loops, what are the advantages of using HOSS? How does the HOSS approach differ from a flat optimization approach is this context?

Another issue concerns the reaction rate vector. Are the reactions limited to zero-order, first-order, or second-order reactions? Are the reaction rates constrained to be the product of reactants with the parameters?

There is a subsection about “Signal back-propagation, reduced Michaelis-Menten mechanisms, and irreversible reactions.” The authors state in line 259, “As signaling pathways models often assume QSS, it is useful to have a tool that reduces mass action models by eliminating complexes.” Michaelis-Menten mechanisms can be important. Can HOSS retain the equation format for these mechanisms? For example, if parameters like ki+, ki-, and kcati are crucial for estimation based on the data, is HOSS capable of estimating these parameters, or does it resort to flat optimization in such cases?

In addition to Michaelis-Menten kinetics, other rate equations, such as Hill functions and sigmoidal functions, can be used to describe reaction rates. Can HOSS incorporate these diverse rate equation formats and estimate the parameters within these equations?

In Figure 3F, it seems that HOSS does not fit the experimental data.

In Figure 6C, it appears that the Flat approach yields a cost value of about 8, while the Hierarchical approach gives a cost value of around 2. However, it’s difficult to determine whether the difference of approximately 6 is significant. The authors should plot the simulation results from both the Flat and Hierarchical approaches alongside the experimental data. This would illustrate what a cost value difference of 6 looks like and clarify whether HOSS offers a real advantage over flat optimization.

In Figure 8D, HOSS optimization is compared to Flat. In D3_EGFR case, Flat performs the same as HOSS, if not slightly better. In D4_EGFR, Flat outperforms HOSS. In D3_b2AR and D4_b2AR, HOSS shows better results than Flat. Based on these outcomes, I don’t see significant benefits in using HOSS over Flat, especially since the improvement in cost value by HOSS is less than 0.05. I’m unsure how meaningful a 0.05 improvement is in this context. As mentioned earlier, the authors should plot the simulation results from both the Flat and HOSS approaches alongside the experimental data. I suspect that a 0.05 improvement in cost value may be indistinguishable when the simulation results are plotted.

The same issues apply to Figure 10B. How significant is an improvement in cost value of less than 0.1 when comparing Flat and HOSS? The authors should plot the simulation results from both the Flat and HOSS approaches alongside the experimental data to demonstrate the importance of a cost value improvement of less than 0.1.

Overall, if the HOSS cannot preserve feedback loops and limits the type of rate functions, but fails to provide a substantial improvement in cost value, it is difficult to justify its advantage over a straightforward flat optimization method.

**Have the authors made all data and (if applicable) computational code underlying the findings in their manuscript fully available?**

Reviewer #1: Yes

Reviewer #2: Yes

Reviewer #3: Yes

PLOS authors have the option to publish the peer review history of their article (what does this mean?). If published, this will include your full peer review and any attached files.

Reviewer #1: No

Reviewer #2: No

Reviewer #3: No
---

## [Decision Letter · Decision Letter 1]

16 Oct 2024

Dear Prof. Bhalla,

Thank you very much for submitting your manuscript "Mathematical basis and toolchain for hierarchical optimization of biochemical networks" for consideration at PLOS Computational Biology.

As with all papers reviewed by the journal, your manuscript was reviewed by members of the editorial board and by several independent reviewers. In light of the reviews (below this email), we would like to invite the resubmission of a significantly-revised version that takes into account the reviewers' comments.

We cannot make any decision about publication until we have seen the revised manuscript and your response to the reviewers' comments. Your revised manuscript is also likely to be sent to reviewers for further evaluation.

Sincerely,

Anders Wallqvist

Academic Editor

PLOS Computational Biology

Marc Birtwistle

Section Editor

PLOS Computational Biology

Reviewer's Responses to Questions

**Comments to the Authors:**

Reviewer #2: The authors have addressed all of my concerns

Reviewer #3: The authors have addressed most of my questions, but two remain unclear to me.

1. The first issue is regarding Figure 3F, which has been changed to Figure 2F in the revised version. In my view, HOSS does not fit the experimental data. The authors explained that “Here we show one plot out of an optimization that included multiple experiments with different profiles, as well as multiple other experiments. The presented plot is one out of a large number of experimental constraints that comprise this parameterization problem, and hence the fit to this plot may have been worsened to improve the overall optimization cost.” However, there are only two plots showing the fittings, Figure 2E and 2F. The authors need to provide the fitting results for several other “experiments with different profiles, as well as multiple other experiments” to support their explanation. I suggest adding about four panels showing different well-fitted protein level changes in addition to aEGFR and aMAPK. The authors need to provide fitting plots demonstrating which proteins are associated with the worsening of aMAPK, rather than vaguely referring to “multiple experiments with different profiles” or “multiple other experiments”. If only two plots are shown, it suggests that HOSS cannot fit both proteins simultaneously.

2. The second issue concerns Figure 7D. This plot confirmed my concern about the meaningfulness of the cost value improvement. In my view, there are no different in the fittings between the plain and hierarchical method for Erk2-pp and Mek1-pp. The hierarchical method might be slightly better than plain in Mos-P, as it avoids the incorrect upward trend. The authors also provided a new Figure 11 showing the distribution of cost values. D3_EGFR and D4 EGFR are nearly identical between HOSS and Flat methods. In the case of D3_b2AR and D4_b2AR, the HOSS method may show an average cost value improvement of 0.1, as shown in Figure 12B. Figure 7D is the only plot comparing the HOSS and Flat methods with the data. Although there may be a 0.1 difference in cost value as shown in Figure 11, Figure 7D demonstrates that this improvement is nearly indistinguishable when the simulation results are plotted. I suggest that author select 3 additional proteins that best highlight the differences between the HOSS and Flat methods in the optimization of reduced MAPK model case. Based on Figure 7D, I would conclude that the cost values for the HOSS optimization are nearly the same as those for the Flat optimization, rather than being better.

Overall, I agree that the advantages of HOSS in execution time and the superiority of modular approaches over direct methods stem from the significant increase in optimization complexity as the problem size grows. However, it is difficult to conclude that the numerical experiments illustrate a gain in precision. Figure 2E and 2F indicate that HOSS by itself cannot fit the data well, and Figure 7D shows that HOSS has not improved precision too much when compared with Flat method. As an optimization tool, I believe it is crucial to clarify how accurately HOSS can fit the data and how much it can reduce the cost values compared to the Flat method.

**Have the authors made all data and (if applicable) computational code underlying the findings in their manuscript fully available?**

Reviewer #2: Yes

Reviewer #3: Yes

PLOS authors have the option to publish the peer review history of their article (what does this mean?). If published, this will include your full peer review and any attached files.

Reviewer #2: No

Reviewer #3: No
---

## [Editor Report · Decision Letter 2]

8 Nov 2024

Dear Prof. Bhalla,

We are pleased to inform you that your manuscript 'Mathematical basis and toolchain for hierarchical optimization of biochemical networks' has been provisionally accepted for publication in PLOS Computational Biology.

Best regards,

Marc Birtwistle

Section Editor

PLOS Computational Biology

Feilim Mac Gabhann

Editor-in-Chief

PLOS Computational Biology

Jason Papin

Editor-in-Chief

PLOS Computational Biology

While some minor disagreements remain about some aspects of the paper, we feel that, given the broad relevance of the topic, it is better to accept and commence with post-publication review, rather than continue with more rounds of peer review.

---

## [Editor Report · Acceptance letter]

22 Nov 2024

PCOMPBIOL-D-24-01317R2 

Mathematical basis and toolchain for hierarchical optimization of biochemical networks

Dear Dr Bhalla,

I am pleased to inform you that your manuscript has been formally accepted for publication in PLOS Computational Biology. Your manuscript is now with our production department and you will be notified of the publication date in due course.

With kind regards,

Lilla Horvath
